# Detection and Localization of F-layer Ionospheric Irregularities with Back Propagation Method Along Radio Occultation Ray Path

Vinícius Ludwig-Barbosa[1], Joel Rasch[2], Thomas Sievert[1], Anders Carlström[2], Mats I. Pettersson[1], Viet Thuy Vu[1], and Jacob Christensen[2]

[1]Blekinge Institute of Technology, Karlskrona, Sweden
[2]Beyond Gravity Sweden AB, Gothenburg, Sweden

**Correspondence:** Vinícius Ludwig-Barbosa (vinicius.ludwig.barbosa@bth.se)

**Abstract.** The back propagation (BP) method consists of diffractive integrals computed over a trajectory path, projecting a signal to different planes. It unwinds the diffraction and multipath, resulting in minimum disturbance on the BP amplitude when the auxiliary plane coincides with the region causing the diffraction. The method has been previously applied in Global Navigation Satellite Systems (GNSS) Radio Occultation (RO) measurements to estimate the location of ionospheric

irregularities but without complementary data to validate the estimation. In this study, the BP method is applied to a Communications/Navigation Outage Forecasting System (C/NOFS) occultation event containing scintillation signatures caused by an equatorial plasma bubble (EPB), which was parameterized with the aid of collocated data and reproduced in wave optics propagator (WOP) simulation. In addition, a few more test cases were designed to assess the BP method regarding the size, intensity and placement of single and multiple irregularity regions. The results show a location estimate accuracy following the

resolution in which the method is implemented (single bubble, reference case), whereas a bias is observed in multiple bubble scenarios. The minimum detectable disturbance level and the estimation accuracy depend on the receiver noise level and, in the case of several bubbles, on the distance between them. These remarks provide insight into the BP results for two Constellation Observing System for Meteorology Ionosphere and Climate (COSMIC) occultation events.

## 1 Introduction

The Huygens-Fresnel method consists of the propagation in a vacuum of a complex wave by computing a diffractive integral of the electromagnetic (EM) field over a plane to one or multiple points in space (Sommerfeld, 1967). The direct form of the line integral is extensively combined with wave optics propagator (WOP) (Knepp, 1983) in order to obtain the EM field equivalent to the Global Navigation Satellite Systems (GNSS) complex signal sampled on the LEO orbit after sounding the Earth's atmosphere during a radio occultation (RO) event (Bevis et al., 1992; Kursinski et al., 1997; Gorbunov and Lauritsen,

2007).

The inverse problem, from Low-Earth Orbit (LEO) orbit towards the GNSS satellite, has been investigated in order to disentangle the multipath and the diffraction from the received total field and to increase the resolution of the bending angle inversion in the lower atmosphere (Gorbunov and Gurvich, 1998a, b; Dahl Mortensen, 1998). The regions with sharp gradients

in refractivity, i.e. non-homogeneities, are the source of diffraction and multipath in amplitude and phase during the forward propagation, according to Huygens' principle (Sommerfeld, 1967). The inverse form of the diffractive integral, hereafter the back propagation (BP) method, computes the projection of the complex signal to BP planes. Ideally, the disturbance observed in the BP amplitude has its lowest on the BP plane placed at irregularity region center. The back propagation field is not fully comparable to the forward field since the back projection is performed in vacuum, i.e. the impact height on the initial plane (boundary condition) is prolonged as straight lines to each BP plane (Gorbunov and Gurvich, 1998a).

The GNSS signal also experiences multipath and diffraction during the ionospheric propagation, where plasma irregularities above $\sim 80$ km altitude are responsible for rapid fluctuations in amplitude and phase, known as ionospheric scintillation (Aarons, 1982; Yeh and Liu, 1982; Wickert et al., 2004). In the E-layer ($\sim 90-130$ km), the regions have enhanced electron density due to the concentration of metallic ions driven by wind shear, magnetic field, gravity waves and the influx of meteors in the Earth's atmosphere; with the main occurrence in mid-latitudes and during summer, and absence around the magnetic equator (Arras and Wickert, 2018; Resende et al., 2018; Yu et al., 2019, 2020; Carmona et al., 2022). In the F-layer, the irregularity regions in low latitudes are commonly referred to as Equatorial Plasma Bubbles (EPB) or Equatorial Spread F (ESF). The phenomenon is driven by the Rayleigh-Taylor instability mechanism with higher occurrence on post-sunset hours (local time), where the recombination of ions in the low altitude creates a vertical gradient in the plasma density extending upwards to the F-region. A natural flow from the less dense (low altitudes) to denser regions (high altitudes) creates depletion areas in the form of plumes (Kelley et al., 1981; Stolle et al., 2006). The higher turbulence and gradient in density on the edges of the up-flowing bubble distorts the EM wave and eventually creates disruption in the operation of radio-frequency systems (Kelly et al., 2014). The irregularities are observed in different scale sizes (Xiong et al., 2016) and the occurrence of EPB has shown significant seasonal, solar cycle and geomagnetic activity dependence (Stolle et al., 2006, 2008; Abdu et al., 2018; Kepkar et al., 2020). In high latitudes, the occurrence of irregularity regions is not restricted in time and mostly originated by particle precipitation triggered by geomagnetic activities besides the global plasma dynamics (Jiao and Morton, 2015; Cherniak and Zakharenkova, 2016).

Following the same principle as in the lower atmosphere, the location of ionospheric irregularities in the E- and F-layer has been estimated with the BP method along the RO ray path (Gorbunov et al., 2002; Sokolovskiy et al., 2002, 2014; Cherniak et al., 2019). Back propagation has been applied to real measurements, but the estimate accuracy has been primarily assessed with WOP simulation of a generic occultation event, including a single iso- or anisotropic irregularity region modelled by one or multiple phase screens (Sokolovskiy et al., 2002). The opportunity of comparing occultation measurements collocated to independent techniques must be taken to evaluate further the capabilities of the BP method in RO measurements. In Carrano et al. (2011), the scintillation pattern observed in a measurement performed by Communications/Navigation Outage Forecasting System (C/NOFS) satellite and caused by a plasma bubble was fully modelled thanks to the parameterization of the disturbance assisted by collocated data, in which the bubble − LEO satellite distance was an important variable. Further, the BP method principle has also been applied to re-scale the scintillation observed in GNSS ground-receiver measurements with different frequencies and to estimate the correlation between the signals (Carrano et al., 2012, 2014).

In this study, the BP method is further assessed with WOP simulations to determine its capabilities and limitations in the context of detection and location of F-layer irregularity regions, i.e., plasma bubbles, in RO measurements. The modelling described in Carrano et al. (2011) is considered as the initial assessment scenario of a plasma bubble in the F-region along the RO ray path, and it was used as a reference to design a few more cases with different placements, sizes, fluctuation intensities and the number of irregularity regions. Sect. 2 introduces the concept of back propagation and its equations in the scenario of an occultation event. Sect. 3 describes the modelling of the ionosphere and plasma bubbles in WOP simulations. Additionally, it addresses the different test cases considered in our evaluation. The simulation results are discussed in Sect. 4 and support the interpretation of two Constellation Observing System for Meteorology Ionosphere and Climate (COSMIC) measurements reported in Cherniak et al. (2019). Finally, the conclusions of the study are summarized in Sect. 5.

## 2   Back propagation

Assuming the scenario of a GNSS-RO simulation, the last stage of a WOP simulation takes place in a region that can be approximated to vacuum. Therefore, the projection of the total field in LEO orbit can be computed by the following diffraction integral (Sommerfeld, 1967),

$$u_o(x,y) = \sqrt{\frac{k}{2\pi}} \int u(x,y) \cos \xi \frac{\exp(ik|\mathbf{r}-\mathbf{r}_o| - i\pi/4)}{|\mathbf{r}-\mathbf{r}_o|^{1/2}} \, dS, \tag{1}$$

where $u$ is the total field at the last phase screen (PS), $k$ is the wavenumber, $\xi$ is the angle between the normal vector to the integration plane ($\hat{N}$) and the segment

$$|\boldsymbol{r} - \boldsymbol{r_o}| = ([\boldsymbol{x_s} - \boldsymbol{x_o}]^2 + [\boldsymbol{y_s} - \boldsymbol{y_o}]^2)^{1/2}, \tag{2}$$

in which the subscripts $s$ and $o$ stand for the coordinates on the phase screen and on the LEO orbit, and $dS$ corresponds to the integration path, i.e., $dy$ in this particular case. Figure 1 shows the RO geometry considered in the computation of the diffraction propagation, where the origin of the coordinate system is the Earth's center.

The total field sampled on the LEO orbit (boundary condition) corresponds to the superposition of a primary and a secondary field. The primary is radiated from the GNSS satellite, whereas the secondary one results from the vibration of ions as the primary field spreads through the ionosphere. The wave field is propagated through sharp gradients in electron density inside the bubble region, which creates nonhomogeneous advances in phase around the F-layer (Culverwell and Healy, 2015). As a result, rapid variations in amplitude and phase will lead to interference in the total field (focusing and defocusing), i.e scintillation (Yeh and Liu, 1982). Figure 2 illustrates the interplay of focusing and defocusing yielded by the electron density gradient, represented in terms of refractive index ($n$), within the irregularity patch and the resultant total field in the observational plane.

By applying the same principles as in the forward propagation, it is possible to propagate the total field sampled along the LEO orbit towards the GNSS satellite. The diffraction integral for the propagation in the opposite direction, known as back

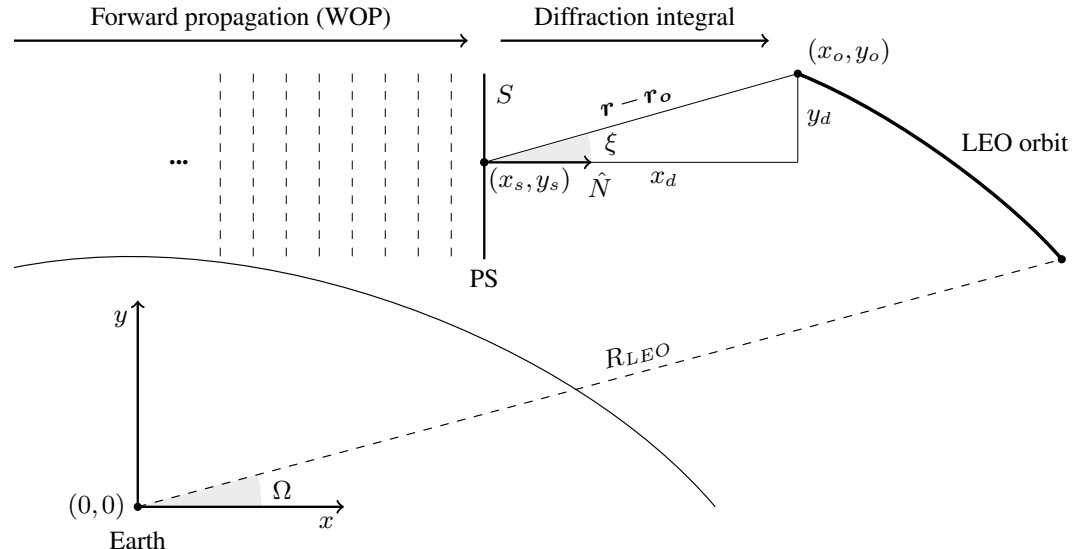

**Figure 1.** Diffraction propagation geometry. In simulations, the forward propagation is modelled with a wave optics propagator (WOP) to consider the ionospheric irregularities. From the rightmost phase screen (PS), the complex signal is integrated along $S$ and projected to every point on the LEO orbit.

propagation method, is written as (Gorbunov et al., 1996; Gorbunov and Gurvich, 1998b; Dahl Mortensen, 1998; Sokolovskiy et al., 2002)

$$u_b(x,y) = \sqrt{\frac{k}{2\pi}} \int u_o(x,y) \cos\xi \frac{\exp(-ik|\mathbf{r}_o - \mathbf{r}_b| + i\pi/4)}{|\mathbf{r}_o - \mathbf{r}_b|^{1/2}} \, dS. \tag{3}$$

However, the integration path $dS$ is not a vertical plane as in the forward direction, but rather $dS = R_{LEO} \, d\Omega$, and the angle $\xi$ is given between the segment $|\boldsymbol{r_o} - \boldsymbol{r_b}|$ and the normal vector to the LEO orbit. The normal vector along the curved path is defined as

$$\hat{\mathbf{N}} = -\cos\Omega\,\hat{x} - \sin\Omega\,\hat{y}. \tag{4}$$

Thus, the final expression for (3) is

$$u_b(x,y) = \sqrt{\frac{k}{2\pi}} \int u_o(x,y) \cos\xi \frac{\exp(-ik|\mathbf{r}_o - \mathbf{r}_b| + i\pi/4)}{|\mathbf{r}_o - \mathbf{r}_b|^{1/2}} R_{LEO} d\Omega, \tag{5}$$

where $\cos\xi = \hat{\mathbf{N}} \cdot \hat{\mathbf{r}}$. Figure 3 shows the geometry of the back propagation scenario, the relation between the angles and the normal vector direction changing along the LEO orbit.

Slightly different procedures are described in the literature to obtain the total field at different BP planes as the complex signal is propagated towards the GNSS satellite:

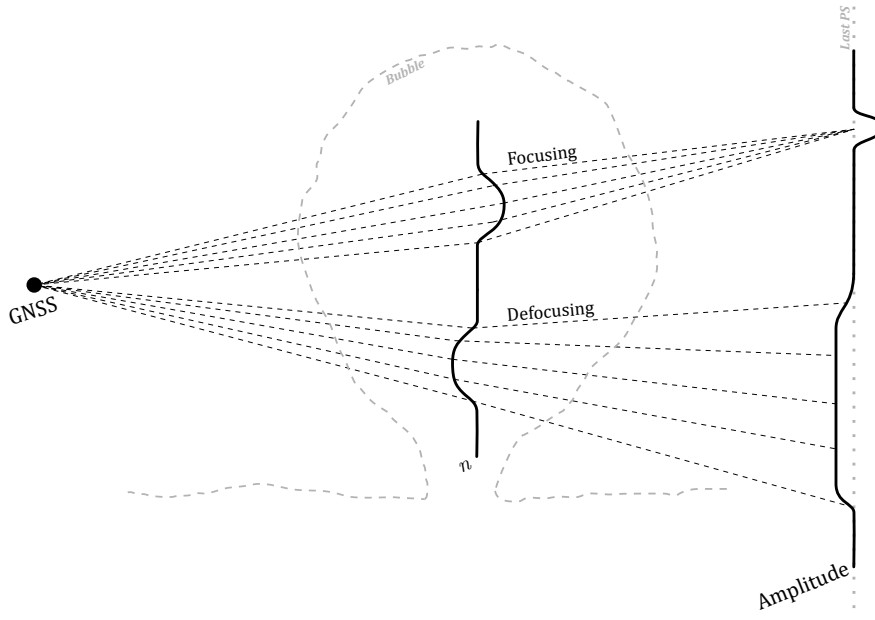

**Figure 2.** Illustration of the wave field generated on the GNSS transmitter (primary field); the ionospheric mechanisms triggered by the secondary field due to the vibration of ions; followed by the overall expansion of the wave field until the observation plane (last phase screen).

1. Compute (5) along the LEO orbit to obtain the BP signal at different BP planes (Sokolovskiy et al., 2002) or

2. Compute the Zverev transform to obtain the BP signal at different BP planes (Gorbunov et al., 2002; Gorbunov and Lauritsen, 2007), which consists of propagating the BP signal at the right-most PS (via diffraction integral) by applying direct and inverse Fourier transforms recursively, viz

$$
\begin{aligned}
\tilde{u}_b(x_b, k_y) &= \mathfrak{F}\left\{u_b(x_b, y_b)\right\}, \quad &(6)\\
u_b(x, y) &= \mathfrak{F}^{-1}\left\{\tilde{u}_b(x_b, k_y)\exp\left(i\sqrt{k^2 - k_y^2}(x_o - x_b)\right)\right\}, \quad &(7)
\end{aligned}
$$

where $\mathfrak{F}$ is the Fourier operator and $k_y$ is the spatial angular frequency.

In order to locate the source of the secondary field, i.e., irregularity regions, the auxiliary plane with minimum BP amplitude disturbance should be found, as the disturbance is expected to reduce gradually up to the source point. The standard deviation of the BP amplitude is the metric used to quantify the disturbance in each auxiliary plane, and it is defined as (Gorbunov et al., 1996),

$$
\sigma_u = \sqrt{\frac{(u' - \overline{u'})^2}{n}}, \quad (8)
$$

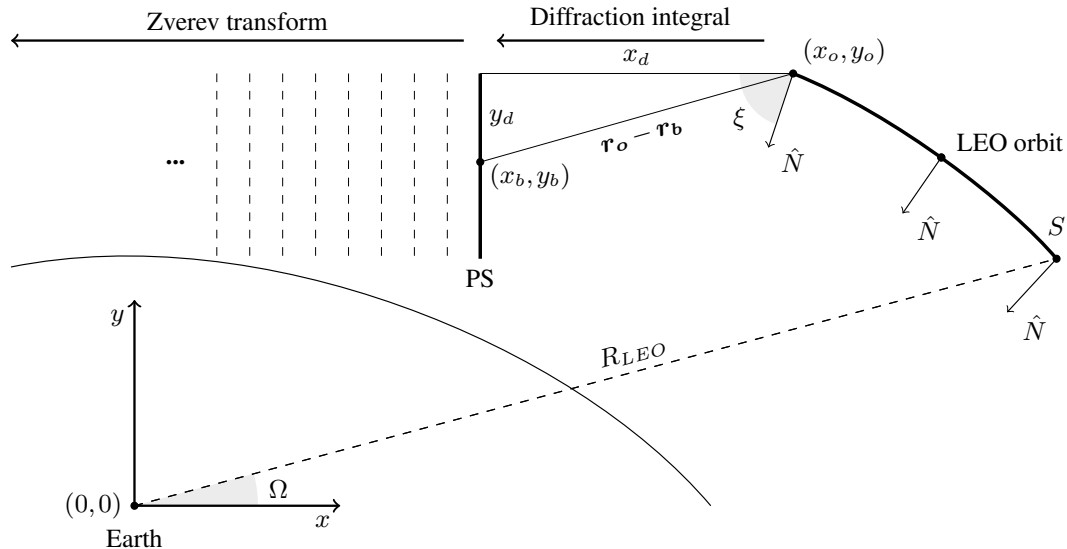

**Figure 3.** Back propagation geometry. The diffraction integral is computed along the LEO orbit path ($S$) and projected to every point on the closest PS. Once the back-propagated signal is available on the auxiliary plane, the Zverev transform is computed iteratively towards the GPS satellite.

where $u' = u - \bar{u}$ corresponds to detrended BP amplitude achieved by a 3-pass $2^{\text{nd}}$ order Savitzy-Golay filter, and assuming a window length of $10\,\text{km}$ – according to the typical irregularity outer-scale (Carrano et al., 2011; Zeng et al., 2019).

The ability to find the origin of the secondary field along the ray path is dependent on the secondary field amplitude (proportional to the electron density gradient), and on the noise level of the LEO receiver. These aspects are investigated in simulations, including the modelling of ionospheric disturbances.

## 3    Ionospheric simulation

The effects of ionospheric refractivity are accounted for in a WOP simulation by assuming the electron density profile (EDP)
as part of the atmospheric model. The refractive index, combining the neutral atmosphere and the ionosphere, is defined as

$$n_i = -40.3 \frac{\rho}{f^2}, \tag{9}$$

$$n = n_n + n_i, \tag{10}$$

where $f$ is the carrier frequency, $\rho$ is the electron density ($\text{el/m}^3$), and subscripts $n$ and $i$ denote the neutral atmosphere and ionosphere, respectively. The addition of the ionospheric model includes the respective phase shift into the total phase
accumulated during the forward wave propagation. From the RO perspective, the excess path due to the ionospheric propagation under such a scenario may result in an extra accumulated bending angle proportional to $f^{-2}$, i.e., the lower the frequency, the

larger the bending. Additionally, the signals in different frequencies have different bending angles due to slightly different propagation paths. Consequently, the signals have different integrated electron densities (Culverwell and Healy, 2015).

## 3.1 F-region irregularity: Plasma bubbles

Under low ionospheric activity, EDPs tend to resemble a slow function (Culverwell and Healy, 2015). Under high activity periods and during the transition between day- and night-time, there is a higher incidence of regions of localized irregularities, e.g., plasma bubbles, leading to a sharper gradient in electron density (Jiao and Morton, 2015; Kepkar et al., 2020). Such regions are responsible for large, medium, and small-scale irregularities, corresponding to sizes up to the Fresnel scale (Xiong et al., 2016). In a RO geometry and especially in the range of ionospheric altitudes where the bending is significantly smaller than in neutral atmosphere (Kursinski et al., 1997), the Fresnel scale is given by

$$d_F = 2\sqrt{\frac{\lambda L_t L_r}{L_t + L_r}}, \tag{11}$$

$$d_F \approx 1.5\,\text{km}, \tag{12}$$

where $\lambda$ is L1 band wavelength, $L_t$ is the horizontal distance of the GNSS satellite to the Earth's limb ($\sim 28.5 \times 10^6$ m), $L_r$ is the LEO horizontal distance ($\sim 3.4 \times 10^6$ m, assuming an altitude orbit of 820 km). The propagation through these irregularities results in diffraction and refraction of the electromagnetic field. These effects are observed as abrupt fluctuations in amplitude and phase, referred to as scintillations (Aarons, 1982; Yeh and Liu, 1982; Wickert et al., 2004; Zeng and Sokolovskiy, 2010). Moreover, the presence of plasma bubbles introduces asymmetries between the inbound (GNSS to tangent point) and outbound (tangent point to LEO) segments of the ray trajectories. This condition contradicts the assumption of spherical symmetry of the atmosphere in retrievals via Abel transform (Fjeldbo et al., 1971), and it is related to high-order terms composing the bias after the standard ionospheric correction (Vorob'ev and Krasil'nikova, 1994). The high-order bias, critical in meteorological and climate applications, is handled either by Kappa or Bi-local correction (Healy and Culverwell, 2015; Liu et al., 2020).

### 3.1.1 Single bubble

The location estimation of the plasma bubbles in the F-region is a complicated task in RO measurements. The ray path between GNSS and LEO satellites includes ionospheric propagation in two segments, i.e., ray inbound and outbound. The disturbance observed in the sampled signal and originated during either the former or the latter segment cannot be visually distinguished. The back propagation (BP) method has been used to detect irregularities in the F-region in studies using both simulations and real occultation measurements (Sokolovskiy et al., 2002, 2014; Cherniak et al., 2019).

However, there is a lack of RO events combined with collocated data provided by different systems where the true location of the irregularity region is precisely known. In this study, the model of isotropic irregularities representing a plasma bubble in the equatorial region is considered in WOP simulations to evaluate the estimation obtained with the BP method. The model has been described in Carrano et al. (2011) and corresponds to a measurement performed by C/NOFS satellite and collocated with an incoherent scatter radar and a Very High Frequency (VHF) ground-based receiver. The collocated data allowed a good

estimation of the placement and size of the bubble, in addition to the parameters required in the modelling of the disturbance observed in the occultation measurement.

The plasma bubble is modelled by a 2-D random realization of Gaussian variables filtered by the spectral density function (SDF),

$$\Phi_{\Delta\rho}(k_x, k_y) = 4\pi\, k_0^{(2\nu-2)} \frac{\Gamma(\nu)}{\Gamma(\nu-1)} \frac{1}{(k_0^2 + k_x^2 + k_y^2)^\nu}, \tag{13}$$

where $k_{x,y}$ are the wave numbers along and across the propagation direction, $k_0 = 2\pi/L_0$ is the outer scale wave number, $\Gamma$ is the Euler's gamma function and $\nu$ denotes the spectral slope. The filtered variables,

$$\Delta\rho(x,y) = \mathfrak{F}^{-1}\left\{ \sqrt{\Phi_{\Delta\rho}(x,y)\, SF}\, r_m \right\}, \tag{14}$$

are modulated to the electron density model,

$$\rho = \rho_b \left[ 1 + \Delta\rho \times \sigma_{\Delta\rho/\rho} \times B \right], \tag{15}$$

where $\rho_b$ is the background EDP and $\sigma_{\Delta\rho/\rho}$ is the root mean square (RMS) level of the fluctuations. The bubble width is a Gaussian envelope function,

$$B(x,y) = e^{\frac{[\alpha(x,y)-\alpha_0]^2}{2\sigma_\alpha^2}}, \tag{16}$$

in which the function maximum and the bell width are set by

$$\alpha_0 = \tan^{-1}\left( \frac{x_0}{hmF2 + R_e} \right), \tag{17}$$

$$\sigma_\alpha = \frac{L_H}{A(hmF2 + R_e)}, \tag{18}$$

where $x_0$ denotes the bubble placement along $x$-axis, $hmF2$ is the F-region electron density peak height, $L_H$ corresponds to
175 the bubble width, $R_e$ is the Earth's radius and the scaling factor $A \approx 1.348$. In (14), $r_m$ corresponds to the grid of Gaussian random numbers, and $SF = L/2\pi$ is a spatial factor in which $L$ is the bubble vertical extension.

The set of parameters estimated in Carrano et al. (2011) was used in our WOP simulation to replicate the scintillation in the total field with equivalent deterministic properties. Further details about the implementation of the wave optics propagator used in the simulations are given in Ludwig-Barbosa et al. (2020); Ludwig-Barbosa et al. (2020). Figure 4 shows the Gaussian
envelope and the filtered random realization modulated to the electron density model.

Figure 5 shows the normalized signal intensity at the observational plane and the power spectral density (PSD) computed within 280 km and 340 km. The results have good agreement with the ones reported in Carrano et al. (2011) and validate our WOP simulation.

The simulated total field disturbed by the plasma bubble during the forward propagation, and sampled at the right-most PS
(see Fig. 1), is considered as the boundary condition to the BP method. The scenario is used as the reference model for different test cases to assess the capabilities and limitations of the BP method in the presence of a single plasma bubble, namely:

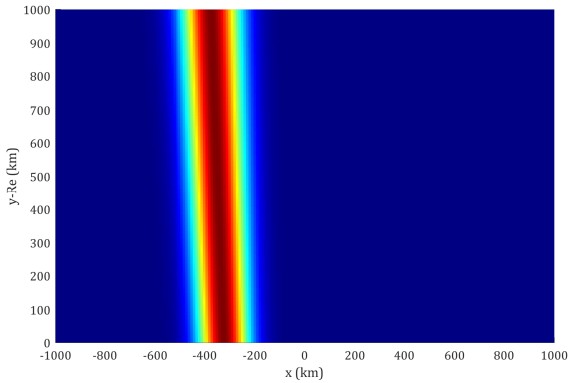
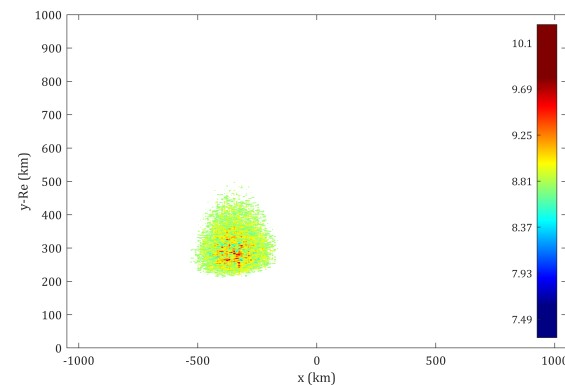

(a) Gaussian function defining the placement and extension along $x$-axis at $x_0 \approx -342.8$ km and assuming $L_H = 102$ km.

(b) Irregularities modulated to an electron density profile (EDP), $hmF2 = 288.5$ km. Color bar unit: $10^{11}$ el/m$^3$.

**Figure 4.** Modelling of the irregularity region used during the forward propagation in simulations. (a) Gaussian envelope defined by (16), (b) resultant irregularities modulated to a background EDP as given in (15).

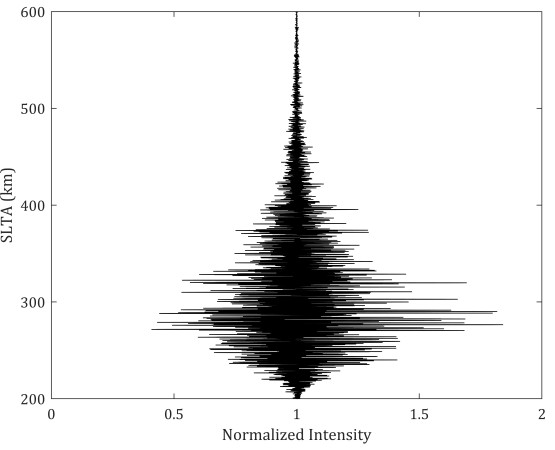
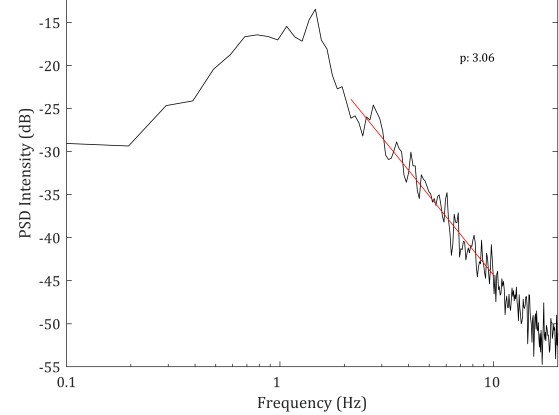

(a) Normalized intensity at rightmost phase screen (observational plane).

(b) Intensity PSD computed in the interval within 280 km and 340 km SLTA. The phase spectral index $p = 2\nu$ refers to the PSD slope estimated in log-log domain (red line).

**Figure 5.** WOP results considering the set of parameters described in Carrano et al. (2011). The original C/NOFS measurement had an average SNR level of around 1500 V/V.

– Accuracy of the location estimate along $x$-axis: the position of the region of irregularities is controlled by modifying $x_0$ in (16);

- Accuracy of the location estimate along $x$-axis with different RMS fluctuation levels: the level of irregularities modulated to the EDP is defined by $\sigma_{\Delta\rho/\rho}$ in (15);

- Accuracy of the location estimate along $x$-axis with different vertical extensions of the bubble;

- Accuracy of the location estimate along $x$-axis with different bubble width: the extension along $x$-axis is controlled by $L_H$ in (18).

### 3.1.2 Multiple bubbles

In addition to the single bubble cases, a second plasma bubble was added to the ray trajectory by superposing another envelope function to the one shown in Fig. 4(a), simply assuming a different $x_0$ in (16). The test cases with two plasma bubbles allow us to evaluate the BP method under the following scenarios:

- Accuracy of the location estimate along $x$-axis for the two plasma bubbles;

- Accuracy of the location estimate to different separation distances between bubbles;

- Accuracy of the location estimate when bubbles have different RMS fluctuation levels.

## 4    Results

In the simulations, the filtered random field was modulated with an EDP modelled by Chapman's function (Culverwell and Healy, 2015) considering the F-region peak ($nmF2 = 8.81 \times 10^{11}\,\mathrm{el/m^3}$), height ($hmF2 = 288.5\,\mathrm{km}$) and scale height ($H = 31\,\mathrm{km}$) according to the EDP described in Carrano et al. (2011).

The forward propagation simulations of the test cases did not include the propagation to LEO orbit via the diffraction integral (1). Therefore, the BP signals are computed via (6,7) since the boundary condition is given on the vertical plane. The WOP signals include instrument noise, which assumed a Meteorological Operational satellite (MetOp-A) occultation event in low latitude as a reference to the signal-to-noise (SNR) level (see Appendix A1). This measurement extends up to 600 km straight-line tangent altitude (SLTA), an exceptional feature compared to nominal MetOp measurements. Normally, the GNSS signal is tracked up to around 100 km SLTA, but an experimental campaign during MetOp-A end-of-life operation had its tracked SLTA range extended to the point where the F-region is included. Different from the neutral atmosphere region, the SNR level decays with altitude due to the antenna radiation pattern. At this particular measurement (and in simulations), the SNR reference level assumed to estimate the instrumental noise strength in the F-region peak was around 600 V/V.

In forward propagation simulations, the closest phase screen to $x_0$ defining the center of the irregularity region was placed at -346.7 km. Therefore, this was the placement reference ($x_{\mathrm{ref}}$) assumed in the accuracy analysis. The BP planes were computed at every 5 km, which defines the precision of the estimations in our implementation.

## 4.1 Single bubble

Fig. 6 shows the BP amplitudes when bubbles were placed at $x_{\text{ref}} = -346.7\,\text{km}$ and assumed RMS fluctuation level $\sigma_{\Delta\rho/\rho} = 17\%$ (Carrano et al., 2011). The plasma bubble is represented in the background of the BP amplitudes (black vertical lines).

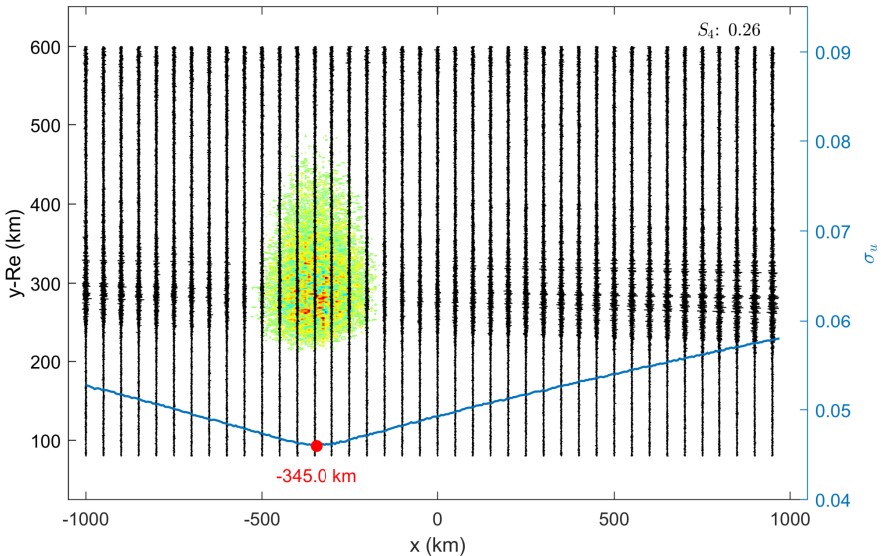

**Figure 6.** Scenario with single F-layer bubble in the inbound region. Irregularity region center around $x_{\text{ref}} = -346.7\,\text{km}$, and assuming $\sigma_{\Delta\rho/\rho} = 17\%$. Black vertical lines represent the detrended BP amplitudes computed in each auxiliary plane (50-km resolution representation). The blue curve corresponds to the detrended BP amplitude standard deviation, computed at every 5 km. The minimum $\sigma_u$ provides the estimated center of the irregularity patch. Estimation error $\varepsilon_x = -1.7\,\text{km}$.

The RMS fluctuation level corresponds to a variation of $\sigma_{\Delta\rho/\rho} \approx \pm 1.5 \times 10^{11}\,\text{el/m}^3$, which results in weak scattering ($S_4 = 0.26$) in agreement to Carrano et al. (2011). The estimate error corresponds, i.e., $\varepsilon_x = x_{\text{ref}} - x_{\sigma_u,\text{min}} = -1.7\,\text{km}$.

Fig. 7 shows the result considering the bubble placement on the ray path outbound.

In the single bubble scenario, the location estimate has good accuracy regardless of the placement in the inbound or outbound sector. Therefore, the location estimate in a single bubble scenario is limited by the precision considered in the BP method,
herein 5 km. The minor difference in the scintillation index ($S_4$) is related to the filtered random variables assumed in the bubble model, which can create a variation in the resultant electron density obtained in the simulation.

### 4.1.1 Influence of RMS fluctuation level

A parametric evaluation of $\sigma_{\Delta\rho/\rho}$ was performed to assess the minimum fluctuation level in which the bubble is detectable with the BP method. Figure 8 shows the box chart comparing the sensitivity in detection for the three different levels, $\sigma_{\Delta\rho/\rho} =$

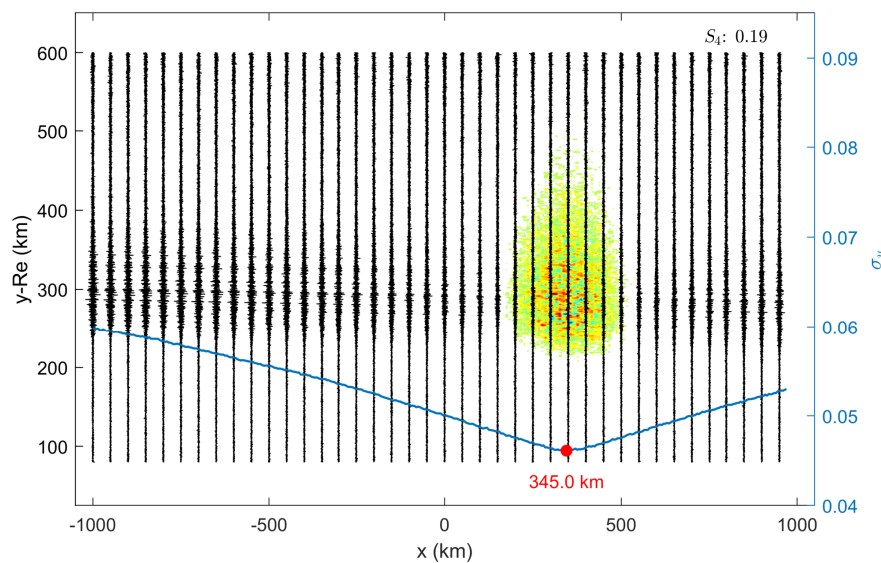

**Figure 7.** Scenario with single F-layer bubble in the outbound region. Irregularity region centered around $x_{\mathrm{ref}} = 346.7\,\mathrm{km}$, and assuming $\sigma_{\Delta\rho/\rho} = 17\%$. Estimation error $\varepsilon_x = 1.7\,\mathrm{km}$.

$2\%, 3.0\%$ and $17\%$ (reference case) in terms of estimation accuracy along the x-axis, and the correspondent BP standard deviation curves.

The curve corresponding to $\sigma_{\Delta\rho/\rho} \leq 2\%$ (red curve in Fig. 8b) does not have a clear global minimum. The BP standard deviation level lies beneath the threshold value determined by the receiver noise level around $hmF2$ ($\sigma_0 \approx 0.0456$ after Fig. 8b and Fig. A1b). Thus, this indicates that the estimations are unreliable when $\sigma_u \leq \sigma_0$. For RMS fluctuation levels $\sigma_{\Delta\rho/\rho} > 3.0\%$

($\pm 2.64 \times 10^{10}$ el/m$^3$), the region of irregularities are detectable with median $x = -327.5\,\mathrm{km}$ and the interval $[-410, -200]\,\mathrm{km}$ corresponding to $50\%$ of estimates ($63.3 < \varepsilon_x < -146.7\,\mathrm{km}$). Regardless, the disturbance level $\sigma_{\Delta\rho/\rho} = 3.0\%$ assumed in the forward propagation corresponds to a weal disturbance at LEO's orbit ($S_4 < 0.1$). At this level of scintillation, the disturbance created by ionospheric irregularities cannot be distinguished from other sources of error. Therefore, the low accuracy achieved in the estimation is not a concerning result. (Béniguel et al., 2009; Ma et al., 2019). For the reference case, an accuracy around

the method precision was achieved ($\varepsilon_x = -3.3\,\mathrm{km}$).

### 4.1.2   Influence of bubble vertical extension

Figure 9 shows the comparison between the location estimate obtained with WOP simulations assuming different vertical thickness for the irregularity region and $\sigma_{\Delta\rho/\rho} = 17\%$. The thickness was controlled by applying a Tukey window to the right term in (15).

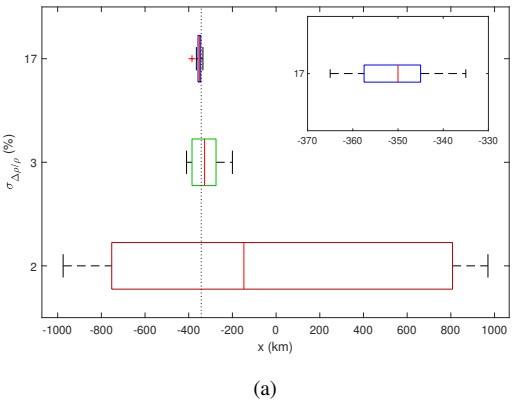 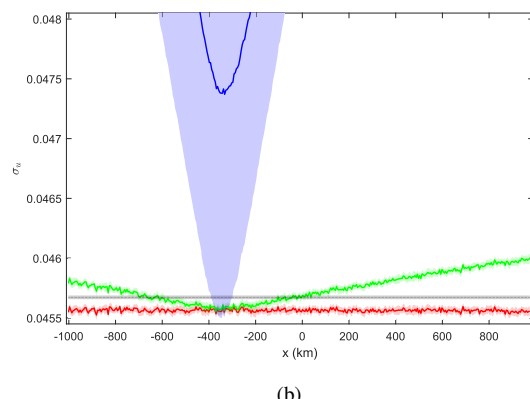

(a)                                                      (b)

**Figure 8.** Influence of fluctuation level in simulations assuming a single irregularity region in the inbound region. (a) Box chart comparing different RMS fluctuation levels. The vertical dashed line indicates the bubble placement. The detection improves significantly when the fluctuations level is $\sigma_{\Delta\rho/\rho} > 3.0\%$ ($\pm2.64 \times 10^{10}\,\mathrm{el/m}^3$), with an estimate median $\bar{x} = -327.5\,\mathrm{km}$ achieved under such set-up. Weaker irregularities, e.g., $\sigma_{\Delta\rho/\rho} = 2\%$, are not distinguishable from the receiver noise and yield poor location estimate of the irregularity patch. (b) BP amplitude standard deviation. Shade regions depict the $2\sigma$-interval. The same color scheme is used in both figures and the grey line represents the receiver noise level. Figures depict results assuming 20 realizations for each fluctuation level.

The black dashed curve shows the standard deviation curve for the bubble with original dimensions, in which the effective extension of the bubbles is defined by the region around the F-region with electron density within 75% of the peak value ($\sim 60\,\mathrm{km}$) (Carrano et al., 2011). The maximum estimation error observed in simulations was -1.7 km for all cases. This result implies the vertical extension of the region does not impact the location estimate.

Despite the disturbance in the simulation being located in F-region, the vertical extensions shorter than 10 km resemble
the thickness of a sporadic E-layer (Zeng and Sokolovskiy, 2010; Arras and Wickert, 2018), and it confirms the capability presented in (Gorbunov et al., 2002; Sokolovskiy et al., 2014; Cherniak et al., 2019). Moreover, the scintillation in E-layer may have a potential advantage in the purview of accuracy given the higher SNR level (lower noise floor) around 100 km (see Fig. A1(b)).

### 4.1.3   Influence of bubble width

Figure 10 shows results for scenarios assuming different bubble widths and fixed fluctuation levels ($\sigma_{\Delta\rho/\rho} = 17\%$). A region with extension $L_H \leq 20\,\mathrm{km}$ creates low scintillation in the GNSS signal ($S_4 < 0.2$), but it is still detectable and it has estimation error $\varepsilon_x = -5\,\mathrm{km}$. Narrower regions do not show a clear global minimum, since the disturbances are at the same level as the receiver noise.

The detection of irregularities is theoretically possible even for wide regions, which leads to higher disturbances as indicated
by the scintillation index. However, the uncertainty about its center estimate increases proportionally to the region width,

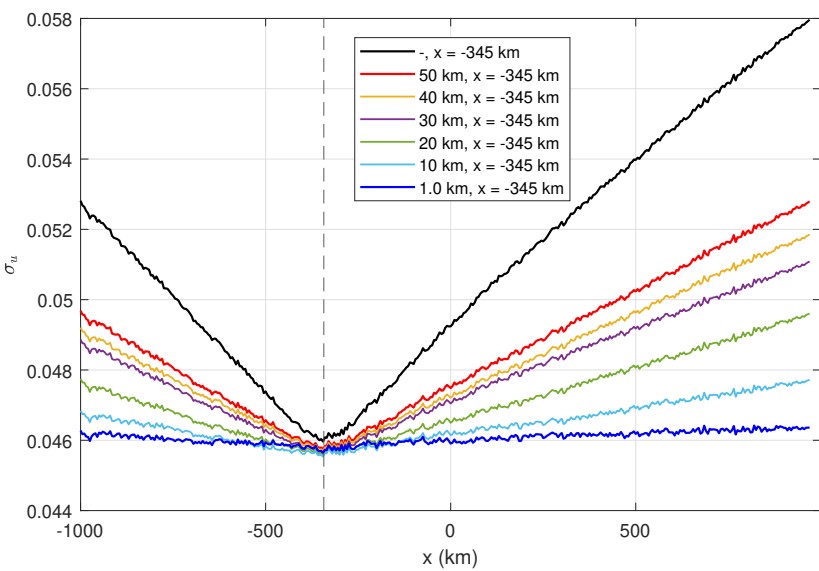

**Figure 9.** BP amplitude standard deviation in scenarios assuming different vertical extensions of a single bubble in the inbound region and $\sigma_{\Delta\rho/\rho} = 17\%$. The black dashed curve corresponds to the reference case. Location estimate is possible up to the thinnest layer, resembling sporadic E-layer dimension. Estimation error, $\sigma_x = -1.7\,\text{km}$. Vertical dashed lines indicate the placement of the irregularity patch.

despite the increasing difference between the global minimum level and the noise floor. Thus, the extension of the irregularity region must be shorter than the distance between GNSS and LEO satellites, as stated in (Sokolovskiy et al., 2002).

## 4.2 Multiple bubbles

Fig. 11 shows two bubbles symmetrically placed around the origin at $x_{\text{ref},1} = -346.7\,\text{km}$ and $x_{\text{ref},2} = 346.7\,\text{km}$, and with the
same fluctuation level ($\sigma_{\Delta\rho/\rho} = 17\%$). The global minimum $\sigma_u$ corresponds to the bubble placed on the outbound region, the last irregularity region along the ray path (forward propagation). The accuracy of the location estimate is affected significantly by the presence of the inbound bubble and by the instrument noise, yielding an estimation error $\varepsilon_x \approx 71.7\,\text{km}$. The location estimate of the inbound bubble is a rather complicated task since the presence of the outbound bubble shadows its contribution to the total wave field and, therefore, a clear local minimum is not detectable in the BP amplitude standard deviation.

Fig. 12 shows the scenario with a larger separation between the irregularity regions, $\Delta x = 1200\,\text{km}$. The minima become more distinguishable and slightly improve their location estimates. The most accurate estimation is given nevertheless on the outbound bubble ($\varepsilon_x \approx 40\,\text{km}$), with the instrument noise having a partial contribution in the error. Regarding the inbound bubble, there is an indication of the irregularity placement around $x = -500\,\text{km}$, but with the estimation error greater than for the outbound patch.

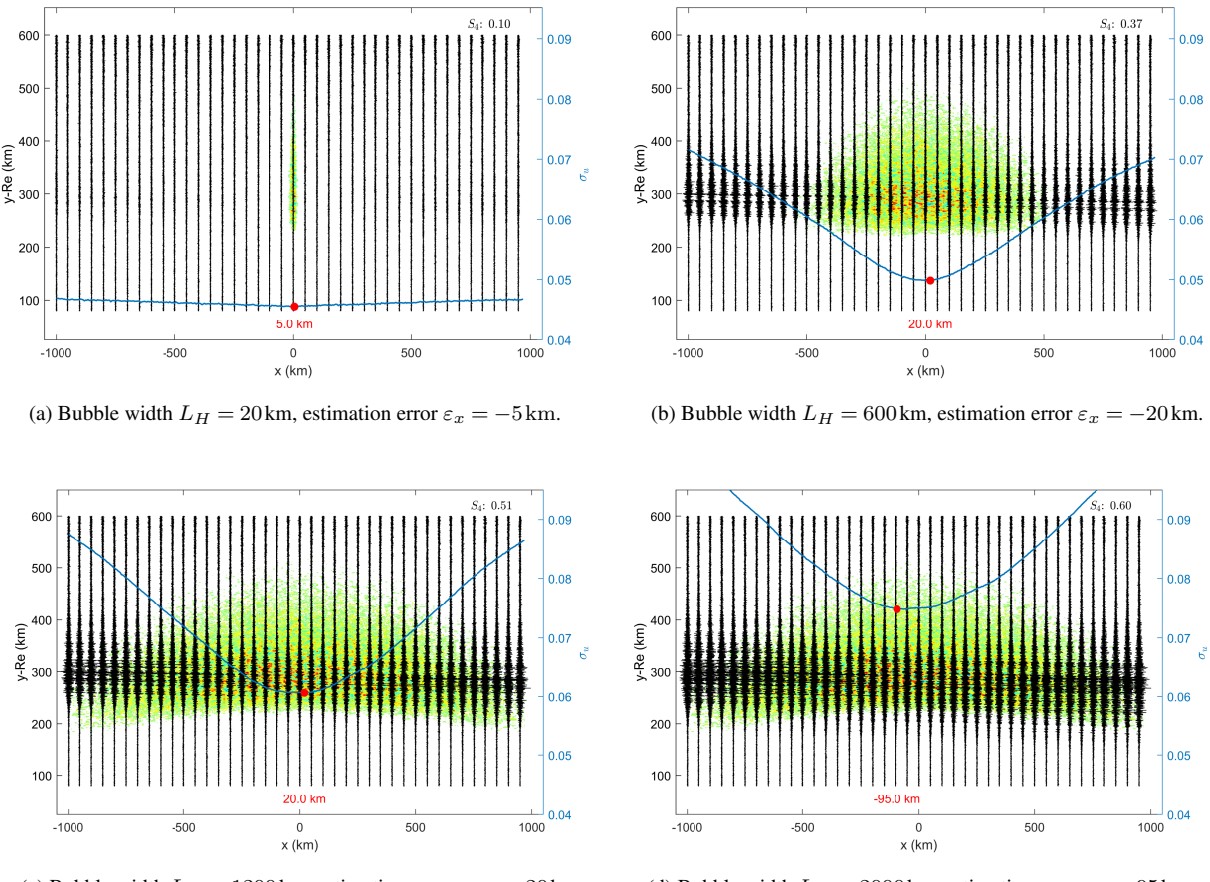

(a) Bubble width $L_H = 20\,\text{km}$, estimation error $\varepsilon_x = -5\,\text{km}$.

(b) Bubble width $L_H = 600\,\text{km}$, estimation error $\varepsilon_x = -20\,\text{km}$.

(c) Bubble width $L_H = 1200\,\text{km}$, estimation error $\varepsilon_x = -20\,\text{km}$.

(d) Bubble width $L_H = 2000\,\text{km}$, estimation error $\varepsilon_x = 95\,\text{km}$.

**Figure 10.** Single bubble with different widths ($L_H$). The detection is possible for wide regions, but estimate accuracy decreases with increasing width.

A comparison between Figs. 10(c,d) (wide bubble scenarios) and Figs. 11 and 12 show that it is possible to distinguish cases with a single wide irregularity region from a scenario with multiple smaller bubbles since the latter would likely present more than one local minimum along the ray path. Nevertheless, the location estimates of secondary patches are less reliable.

In contrast to single region cases where the predominant constraint to detection is the noise level ($\sigma_u \approx \sigma_0$), these results indicate that the separation between the regions has a major influence on the detection/location task of multiple patches.

### 4.2.1 Influence of RMS fluctuation level

In these test cases, the RMS fluctuation level of one of the bubbles was kept constant ($\sigma_{\Delta\rho/\rho} = 17\%$) while the other had the fluctuation set to weaker values. Fig. 13 depicts the results assuming $\sigma_{\Delta\rho/\rho} = 6\%$ and $\sigma_{\Delta\rho/\rho} = 12\%$.

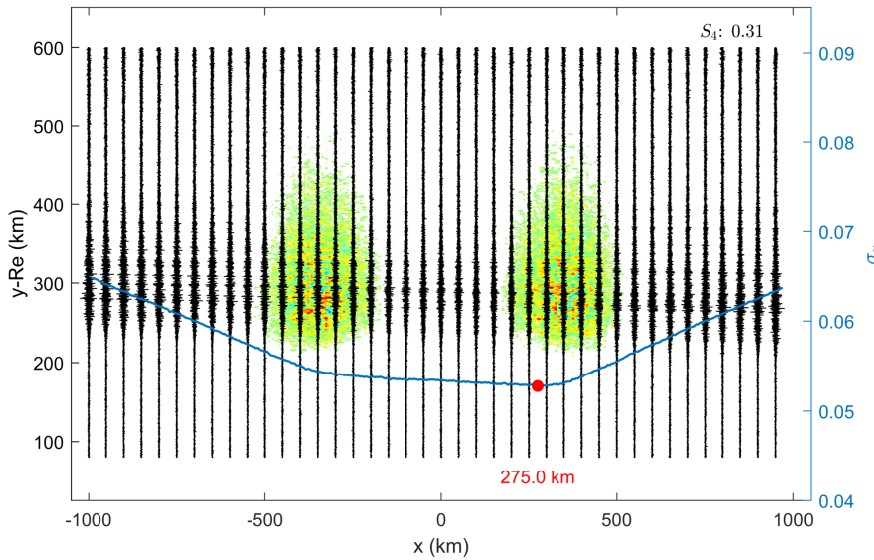

**Figure 11.** Bubbles at $x_{\mathrm{ref},1} = -346.7\,\mathrm{km}$ and $x_{\mathrm{ref},2} = 346.7\,\mathrm{km}$, $\sigma_{\Delta\rho/\rho} = 17\%$. Estimation error $\varepsilon_{x2} = 71.7\,\mathrm{km}$.

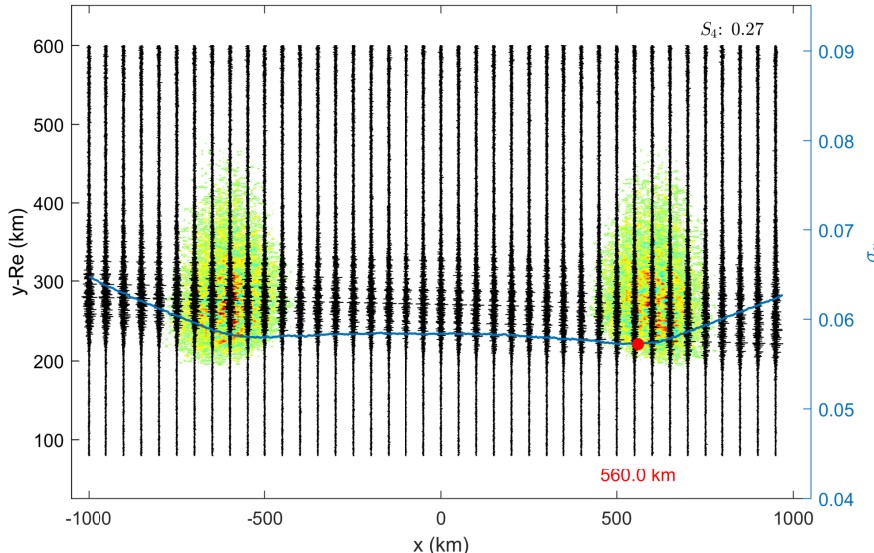

**Figure 12.** Bubbles at $x_1 = -600\,\mathrm{km}$ and $x_2 = 600\,\mathrm{km}$, $\sigma_{\Delta\rho/\rho} = 17\%$. Estimation error $\varepsilon_{x2} = 40\,\mathrm{km}$ and an indication of the inbound bubble's placement along the ray path.

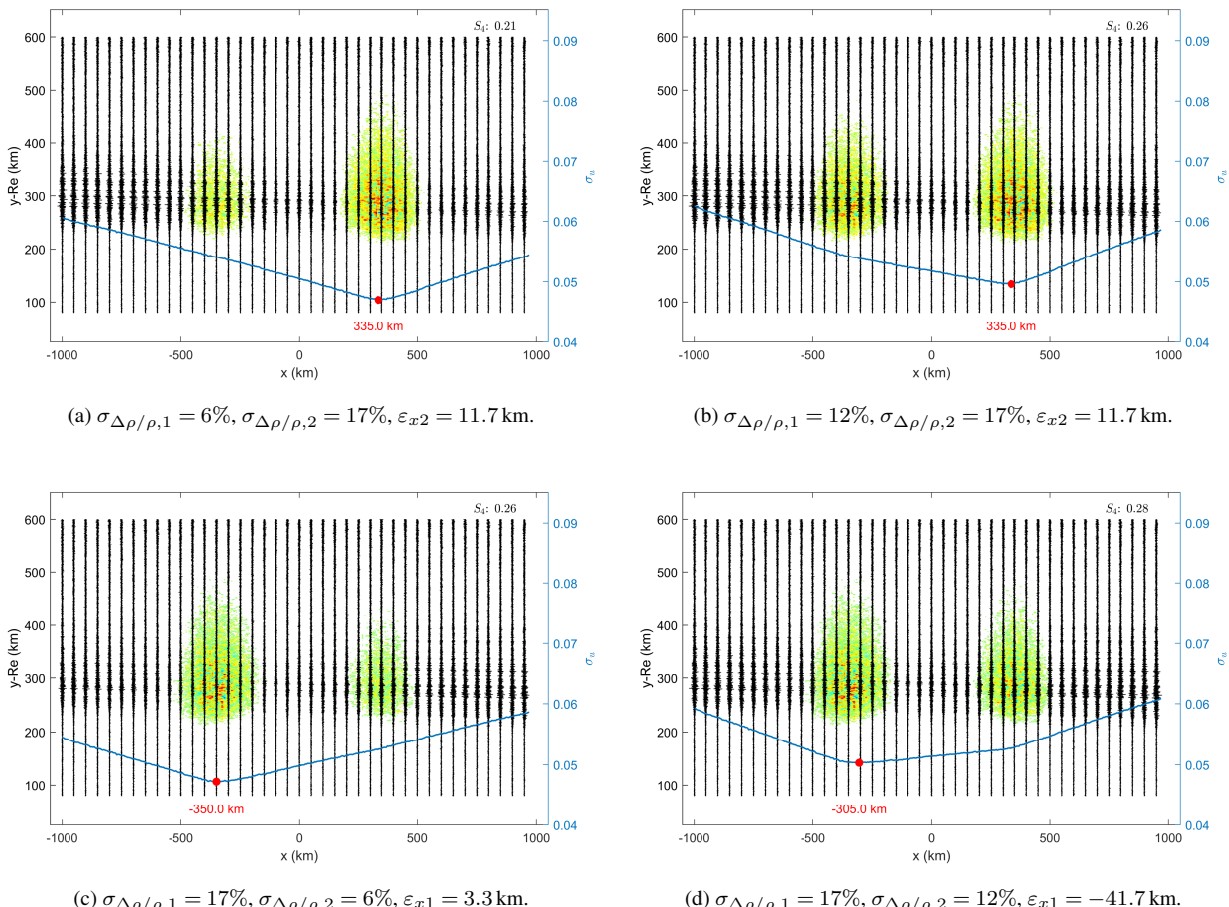

(a) $\sigma_{\Delta\rho/\rho,1} = 6\%$, $\sigma_{\Delta\rho/\rho,2} = 17\%$, $\varepsilon_{x2} = 11.7\,\mathrm{km}$.

(b) $\sigma_{\Delta\rho/\rho,1} = 12\%$, $\sigma_{\Delta\rho/\rho,2} = 17\%$, $\varepsilon_{x2} = 11.7\,\mathrm{km}$.

(c) $\sigma_{\Delta\rho/\rho,1} = 17\%$, $\sigma_{\Delta\rho/\rho,2} = 6\%$, $\varepsilon_{x1} = 3.3\,\mathrm{km}$.

(d) $\sigma_{\Delta\rho/\rho,1} = 17\%$, $\sigma_{\Delta\rho/\rho,2} = 12\%$, $\varepsilon_{x1} = -41.7\,\mathrm{km}$.

**Figure 13.** Bubbles at $x_{\mathrm{ref}} = \pm346.7\,\mathrm{km}$ with weaker RMS fluctuation level on the inbound (a,b) and the outbound bubble (c,d).

The standard deviation curves in scenarios including a bubble with $\sigma_{\Delta\rho/\rho} = 6\%$ are similar to the one observed in the scenario of a single bubble (see Fig. 6 and 7). However, the location of the global minima along the $x$-axis differs, indicating that the presence of a weaker bubble affects the location estimate of the predominant irregularity region. The remarks are valid despite the placement of the weaker region in the inbound or outbound sector. However, the inbound bubbles have a greater impact on the estimation accuracy of the inbound bubbles than the opposite.

Figure 14 shows the comparison of standard deviation curves assuming different RMS fluctuation levels on the bubble placed at the inbound sector. A clear shift of the global minimum towards the weaker patch is observed around $x_{\mathrm{ref}}$ as $\sigma_{\Delta\rho/\rho,1}$ increases from $6\%$ to $17\%$ ($\sigma_{\Delta\rho/\rho,1} = \sigma_{\Delta\rho/\rho,2}$), which leads to a gradual increase in the estimation error. After $\sigma_{\Delta\rho/\rho,1} > \sigma_{\Delta\rho/\rho,2}$, the estimation indicates the position of the inbound bubble and no longer the outbound one.

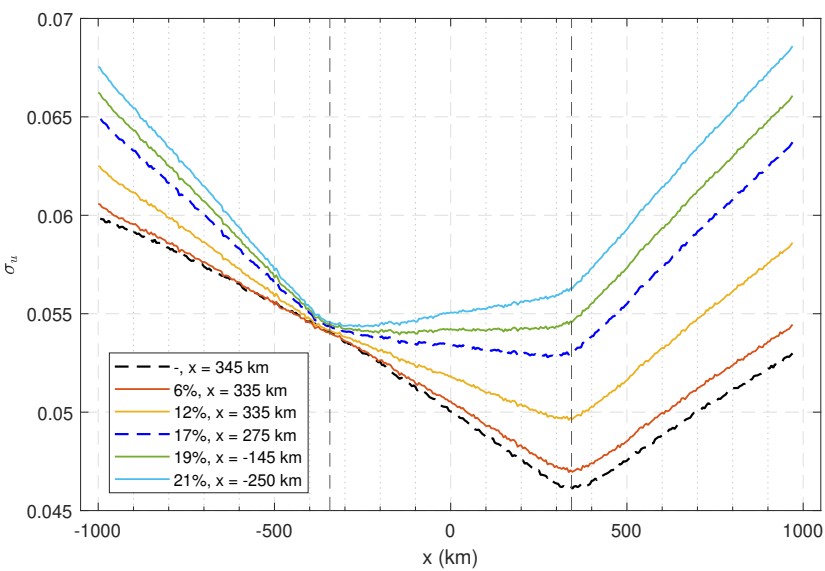

**Figure 14.** Comparison between simulations assuming different RMS fluctuation levels on the inbound bubble and constant on the outbound bubble ($\sigma_{\Delta\rho/\rho,2} = 17\%$). The legend shows $\sigma_{\Delta\rho/\rho,1}$ level and the location estimate along $x$, given after the $\sigma_u$ global minimum. Vertical dashed lines depict the placement of the irregularity regions in the simulations. The black dashed curve corresponds to the case of a single bubble in the outbound sector. The blue dashed curve corresponds to the scenario shown in Fig. 11.

## 4.3 Analysis of COSMIC occultations events

The remarks made after the test cases are used in the evaluation of two COSMIC occultations events presented in Cherniak et al. (2019). The measurements were performed during a severe geomagnetic storm between June $22^{nd} - 23^{rd}$, 2015. Their results are replicated in Fig. 15 after using (5) to compute the BP amplitude at $x = 3000\,\mathrm{km}$, followed by employing (6,7) recursively to obtain the total field at the other auxiliary planes.

The global minima are found between $2600 - 2800$ km in both occultations, indicating the position of the main region of irregularities along the ray path. In Fig. 15(a), the BP amplitude standard deviation was computed assuming the entire height range available in every BP plane since the measurement SNR (figure not shown) did not contain any clear signature of sporadic-E scintillation (Zeng and Sokolovskiy, 2010; Arras and Wickert, 2018; Yu et al., 2020; Carmona et al., 2022). In Fig. 15(b), an u-shape fade was presented around 100 km SLTA (figure not shown). This altitude corresponds to the conventional range of occurrence for sporadic-Es, likely indicating that the irregularities were aligned with the propagation direction. Therefore, the height range around the u-shape fade has not been included in the calculation of the BP amplitude standard deviation and so it does not affect the location estimate of disturbances in the F-region. The same methodology has been assumed in (Gorbunov et al., 2002; Cherniak et al., 2019).

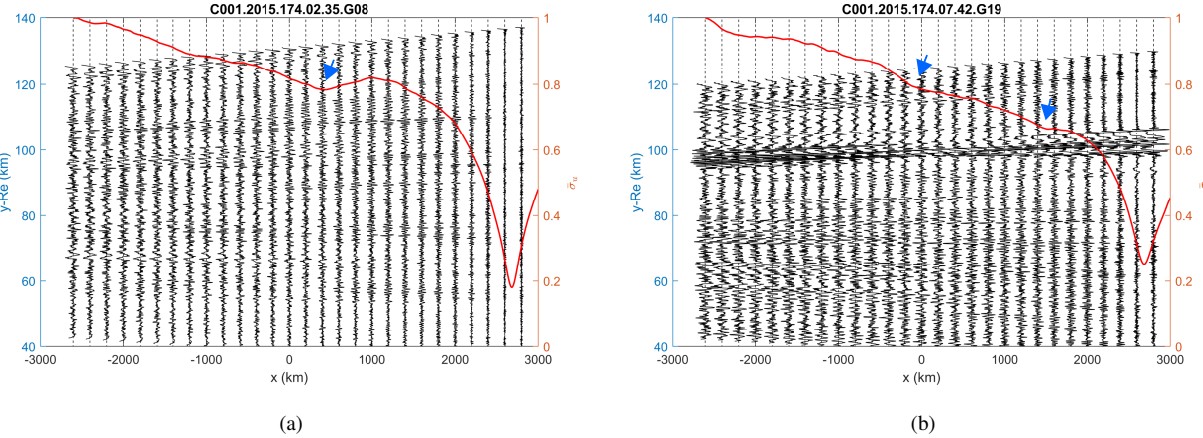

(a)

(b)

**Figure 15.** BP amplitudes of COSMIC occultations events during geomagnetic storms. The right $y$-axis corresponds to the values of the normalized BP amplitude standard deviation (red curve). The global minimum in the red curves estimates a similar position of the main irregularity region for both occultations ($x = 2600 - 2800$ km). The arrows highlight local minima, which may indicate the existence of other regions of ionospheric irregularities. Measurement average SNR level (a) 564 V/V, (b) 694 V/V.

As seen in simulations, the existence of other local minima in the standard deviation curve gives the indication of not only one but two or more irregularity regions during the occultation events. The arrows point out the approximate location of these local minima in Fig. 15. The confirmation of the existence of such regions requires collocated measurements, similar to the case reported in Carrano et al. (2011). In addition, the existence of multiple regions has been shown to reduce the estimate accuracy given after the global minimum to some extent. In reality, the main irregularity region could have been placed slightly closer to the LEO satellite than the position estimated by the BP method, whereas the secondary patches may be slightly farther away from the receiver (similar to Fig. 12).

## 5 Conclusions

The capability of back propagation to detect irregularity regions in the F-layer, e.g., ionospheric plasma bubbles, has been assessed with WOP simulations. The reference case corresponded to a single bubble at the inbound sector observed in a C/NOFS occultation event, in which the location, size, and distance from LEO orbit have been confirmed with collocated data (Carrano et al., 2011). The same model of isotropic irregularities was applied to all the other test scenarios evaluated with WOP simulations.

In the simulation of single bubble scenarios, the location estimate accuracy of the irregularity region along the ray path follows the method resolution for the reference case ($\sigma_{\Delta\rho/\rho} = 17\%$). The bubble placement in either inbound or outbound regions did not affect the detection and location estimate of the irregularity regions. Additionally, the detection of bubbles has

been possible regardless of the region width or vertical extension when $S_4 > 0.2$. However, the accuracy of the center estimate decreases with increasing width.

In multiple bubble scenarios, the ability to estimate the location of bubbles requires the patches to be well separated. Then, the regions are detectable, but the accuracy of the estimate differs. The region yielding the stronger disturbance (predominant) has the most accurate location estimate. However, a bias towards the weaker bubbles is inherent, and it increases with the RMS fluctuation level. If secondary bubbles have a very weak fluctuation strength, the patch is shadowed by the dominant region, and their existence can be untraceable. In the case of aligned bubbles with similar intensities, the most accurate estimation corresponds to the latest region along the forward propagation direction.

Most importantly, the capability of detection/location of irregularity patches has shown to be limited by the receiver noise level, i.e., localizing irregularity patches with the BP method is unfeasible when the noise level is greater than the amplitude of the ionospheric scintillation ($\sigma_u < \sigma_0$). At the SNR level assumed as the reference in our simulations (MetOp), even irregularity patches in the F-region corresponding to low scintillation were detectable ($\Delta\rho \sim \pm 2.64 \times 10^{10}\,\mathrm{el/m^3}$). This fluctuation corresponds to the local gradient within the bubble region and, therefore, depends on the local mean density (background EDP), patch size, and distance between the bubble and receiver. Nevertheless, the minimum detectable level will vary among different receivers according to their noise figures.

The SNR levels, as well as the highest SLTA points in measurements, differ in different RO missions. An SLTA range which includes the ionospheric layer, i.e., further than 100 km SLTA as seen in the experimental MetOp-A campaign, is an important feature to accurately detect and locate the ionospheric plasma bubbles in RO measurements. A minimized influence of the antenna gain in higher SLTA might also contribute to improving the results obtained with the BP method. Nonetheless, the results indicate that the present operating SNR level in the MetOp constellation is sufficient to detect even low scintillation levels.

The information about the location of irregularity regions, e.g., plasma bubbles, is relevant in ionospheric modelling and could potentially support mapping such phenomena and their climatology. In this context, RO data has the potential to improve the gaps in the coverage provided by networks of ground-based receivers detecting and tracking these regions. Our results should be taken as a complement to the investigations described in Gorbunov et al. (2002); Sokolovskiy et al. (2002); Cherniak et al. (2019). Further evaluations of collocated occultation events with data provided by different systems, in line with Carrano et al. (2011), are required to evaluate the method capabilities, also regarding E-layers. In combination with the location along $x$, the horizontal and vertical extension of the plasma irregularity are also parameters of great interest to modelling the plasma irregularities. Approaches to estimate such features, as well as alternatives to locate secondary regions, should be investigated in future works.

*Code and data availability.* Codes and measurements are provided under request to the authors.

## Appendix A: Including instrument noise in WOP signals

In WOP simulations, the signal transmitted by the GNSS (boundary condition) is assumed to be a cylindrical wave. The propagation between the GNSS satellite and the first phase screen occurs in free space, with amplitude decay $\propto 1/\sqrt{r}$. For the sake of practicality, the complex signal is normalized on the first PS. Then, the medium refractivity is recursively accounted by modifying the instant phase of the incident wave and propagating it in a vacuum until the next neighbouring phase screen (Knepp, 1983). At the last PS, the normalized complex signal in the WOP ($\hat{u}$) can model a real signal by using a constant calibration factor, $A$, viz

$$u_{signal}(t) = A\,\hat{u}(t). \tag{A1}$$

The total signal will also include noise,

$$u_{total}(t) = u_{signal}(t) + u_{noise}(t). \tag{A2}$$

We used the measured SNR from a representative MetOp-A occultation event to estimate the appropriate noise level added in the WOP amplitude,

$$\hat{u}_{noise}(t) = \frac{1}{A}u_{noise}(t), \tag{A3}$$

$$\hat{u}_{total}(t) = \hat{u}(t) + \hat{u}_{noise}(t). \tag{A4}$$

The noise in occultation measurements has several sources: thermal noise in the receiver; clock noise; co-channel noise. For this task, we assumed a normal distribution to model the white noise, i.e, $X, Y \sim \mathcal{N}(\mu, \sigma^2)$, where $\mu$ is the mean value and $\sigma^2$ is the variance. Then, the noise in the $i$-th sample is

$$u_{noise}(t_i) = \sigma_0(X + jY)/\sqrt{2}, \tag{A5}$$

$$\sigma_0' = \sigma_0/A, \tag{A6}$$

$$\hat{u}_{noise}(t_i) = \sigma_0'(X + jY)/\sqrt{2}, \tag{A7}$$

where $X, Y \sim \mathcal{N}(0, 1)$. Next, we obtain the average noise power (approximation due to the finite number of samples) by multiplying the noise with its complex conjugate and taking the average over a large time window,

$$P_{noise} = \langle u_{noise}\,u_{noise}^* \rangle \approx \sigma_0^2. \tag{A8}$$

Likewise, the averaged signal power becomes

$$P_{signal} = \langle u_{signal}\,u_{signal}^* \rangle \approx A^2 \langle \hat{u}\,\hat{u}^* \rangle. \tag{A9}$$

The SNR in terms of the signal and noise power, with units [W/W], is given by

$$SNR_W = \frac{P_{signal}}{P_{noise}} \approx \frac{A^2 \langle \hat{u}\,\hat{u}^* \rangle}{\sigma_0^2}. \tag{A10}$$

Hence,

$$\sigma_0 = \sqrt{\frac{A^2 \langle \hat{u}\hat{u}^* \rangle}{SNR_W}}, \tag{A11}$$

and

$$\sigma_0' = \sqrt{\frac{\langle \hat{u}\hat{u}^* \rangle}{SNR_W}}. \tag{A12}$$

In case different sample rates are used in the measurements and the simulations, one has to take into account the sample rate
or the bandwidth ($B$), where

$$B \propto f_s, \tag{A13}$$

in which $f_s$ is the sample rate in Hz. The noise power is given by

$$P_{noise} = BN_0, \tag{A14}$$

where $N_0$ is the noise power density in W/Hz, which is assumed to be a distinct constant for each occultation event. Thus, the
390 SNR to be assumed in the simulations is related to the measured SNR as

$$SNR_{W,WOP} = \frac{P_{signal}}{P_{noise,,WOP}} = \frac{P_{signal}}{P_{noise}} \frac{B}{B_{WOP}} = SNR_W \frac{B}{B_{WOP}} = SNR_W \frac{f_s}{f_{s,WOP}}. \tag{A15}$$

Then, the final formula for the noise amplitude to be added to the WOP signal is

$$\sigma_0' \approx \sqrt{\frac{\langle \hat{u}\hat{u}^* \rangle}{SNR_{W,WOP}}} = \sqrt{\frac{\langle \hat{u}\hat{u}^* \rangle}{SNR_W} \frac{B_{WOP}}{B}} = \sqrt{\frac{\langle \hat{u}\hat{u}^* \rangle}{SNR_W} \frac{f_{s,WOP}}{f_s}}. \tag{A16}$$

Conventionally, the SNR is described in terms of voltage ratio in the RO community. In this way,

$$SNR_W[W/W] = SNR_V^2[V/V]. \tag{A17}$$

Finally,

$$\sigma_0' \approx \sqrt{\frac{\langle \hat{u}\hat{u}^* \rangle}{SNR_V^2} \frac{f_{s,WOP}}{f_s}}, \tag{A18}$$

which completes the derivation for the noise signal strength to be added to WOP signals.

The instrument noise added to WOP signals assumed SNR of a MetOp-A occultation event as the reference in (A18).
The measurement is part of an end-of-life experimental campaign performed by EUMETSAT (European Organization for the
Exploitation of Meteorological Satellites), where the vertical coverage of the GRAS instrument was extended temporarily up
to 600 km SLTA (originally $\sim$ 80 km SLTA). Fig. A1 shows L1 C/A SNR of the occultation event scaled to $f_s = 1$ Hz, which
was not affected by ionospheric disturbances ($S_4 \leq 0.2$), and the WOP amplitude with added noise on the last PS.

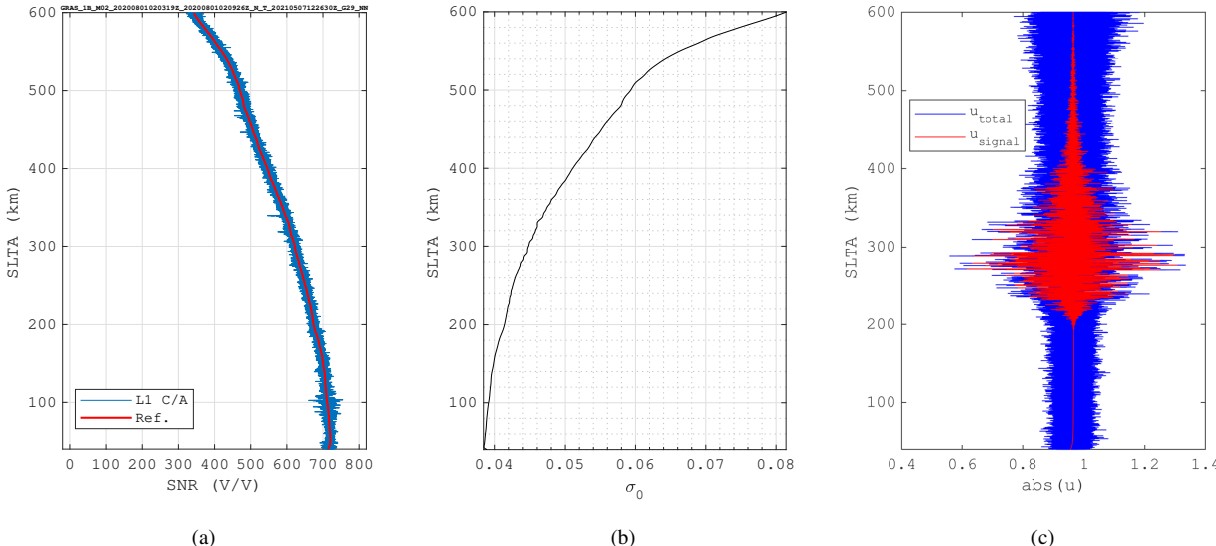

(a)  (b)  (c)

**Figure A1.** (a) L1 C/A $SNR_{1Hz}$: Blue curve shows the original SNR, and the red curve depicts the averaged curve, which values were used as reference in (A18). The decay in SNR observed with increasing SLTA ($> 100$ km) is due to the antenna gain pattern. (b) Amplitude of the noise added to the WOP signal. (c) WOP amplitude with and without added noise on the observational plane (last PS), single inbound bubble scenario (Carrano et al., 2011).

In our WOP simulations, the GNSS signal is propagated up to the rightmost PS. In order to define $f_{s,WOP}$ in this particular scenario, the scanning velocity was approximated to $v_s = 3.2$ km/s. Given the number of points per PS ($2^{18}$) and screen height (1000 km), the WOP sampling frequency in (A18) is $f_{s,WOP} = 839$ Hz.

The $S_4$ index presented throughout the evaluations includes the added instrument noise. Thus (Syndergaard, 2006),

$$S_4 = \frac{\sqrt{\langle (I - \langle \bar{I} \rangle)^2 \rangle}}{\bar{I}}, \tag{A19}$$

where the signal intensity $I \propto |\hat{u}_{total} \hat{u}_{total}^*|$, $\bar{I}$ stands for the filtered intensity and $\langle \rangle$ correspond to 10-s average.

*Author contributions.* VLB, JR and TS designed the study cases and VLB performed the simulations and processing. VLB prepared the manuscript and its revised versions with contributions from all co-authors. JR prepared the Appendix.

*Competing interests.* The authors declare that they have no conflict of interest.

*Acknowledgements.* This research was supported by National Space Engineering Program (NRFP-4), funded by Swedish National Space Agency (Rymdstyrelsen). The authors would like to thank Riccardo Notarpietro (EUMETSAT) for sharing the GRAS ionospheric extension experiment on EPS Metop-A data, which was used as a reference in this work.

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
