# Peer review of "Detection and Localization of F-layer Ionospheric Irregularities with Back Propagation Method Along Radio Occultation Ray Path"

_Atmospheric Measurement Techniques, 2022_

## Author Comment (AC1)

**Editorial Board and Referee #1**
**Atmospheric Measurement Techniques, EGU**

| **Manuscript ID:** | **Correspondent:** |
| --- | --- |
| AMT-2022-57 | Vinícius Ludwig Barbosa |
| | Department of Mathematics and Natural Sciences |
| | Blekinge Institute of Technology |
| | Campus Gräsvik, 371 79 Karlskrona |

**Dear Referee #1,**

Thanks for your comments on AMT-2022-57 "Detection and Localization of F-layer Ionospheric Irregularities with Back Propagation Method Along Radio Occultation Ray Path". The authors have considered your comments and this document contains our answers to them.

An updated manuscript containing modifications based on the comments received during the review process will be submitted after the "open discuss" stage is closed.

Best regards,
Authors

Blekinge Institute of Technology                                    Telephone: +46 736 22 36 31
Department of Mathematics and Natural Sciences (TIMN)                          E-mail: vlb@bth.se
Campus Gräsvik                                                        https://www.bth.se/timn
371 79 Karlskrona

[Figure]

**Referee's comment**

The authors seem to have contradictory understanding about the use of back propagation methods to localize the ionospheric irregularities. On one hand, the authors, as titled, studied the back propagation method to localize F-layer ionospheric irregularities in this manuscript. On the other hand, the authors seem to have deep suspicion about this method. For example, in L137-140, the authors mentioned that "BP method has been used to detect irregularities in F-region in studies using both simulations and real occultation measurements. However, there is a lack of RO events combined with collocated data provided by different systems where the true location of the irregularity region is precisely known". So can the back propagation method localize the irregularities or not? Clearly there are only rare chances we can find RO events collocated with other observations. Also if other observations can provide true location of irregularity, why do we need RO data for this purpose then?

> **Authors' reply:**
>
> We are not suspicious about the method or the concepts involved in it. Our investigation is based on the fact that the references listed in our manuscript, the ones in which BP method is applied to ionospheric irregularities, do not assess the accuracy of such estimations by comparing them to measurements provided by other techniques.
>
> Many published studies compare data provided by two or more techniques. Therefore, we understand there is a scientific value in comparing the estimation obtained with RO events to other sensors. At the current stage, we have used one particular case with multiple collocated measurements. Future investigations should use more combined events (as many as possible) to show how accurate the estimation obtained with RO data can be in different scenarios.
>
> The location of irregularity regions using RO data is important because it can improve the coverage provided by ground-based receivers, especially over water and in remote areas. This statement will be included in the next version of the manuscript.

**Referee's comment**

The study took an RO case used in Carrano *et al.* (2011) as a starting point to perform their simulations. The authors thought the location of the plasma bubble in this case can be well estimated because it collocates with observations from a radar and a ground-based VHF receiver. I agree that from different observation platforms you can derive more physical parameters related to the irregularity. But one reason why Carrano *et al.* (2011) didn't use the back propagation method to infer the location of irregularity was the phase data (which is required to back propagate the signal) was not available for this case.
* * *
Blekinge Institute of Technology
Department of Mathematics and Natural Sciences (TIMN)
Campus Gräsvik
371 79 Karlskrona

Telephone: +46 736 22 36 31
E-mail: vlb@bth.se
https://www.bth.se/timn

[Figure]

**Authors' reply:**
We were unaware of this limitation faced by Carrano *et al.* (2011). Nevertheless, they presented a case of a plasma bubble affecting an occultation measurement, which was modelled in MPS simulation. Their amplitude comparison between measurement and simulation showed quite good agreement. Our replication of such a result is shown in Fig. 5. Therefore, the simulated complex wave is available to perform the back propagation. The results obtained in this case led to the investigation of the other test scenarios.

**Referee's comment**
In my opinion, the back propagation method is not something new. It has been used to localize the ionospheric irregularities in simulation studies and real RO measurements.

- The simulation of the single bubble case in this study was similar/same to the modeling described in Carrano *et al.* (2011). What is new in the study is that more cases with different sizes, fluctuation intensities and placements were designed to test the impact on the estimation accuracy. But I am not sure how applicable of those designed simple cases is to represent the real ionospheric irregularities, and not very convinced by the conclusions made through such simple analysis. The authors claimed that the location estimation accuracy of the back propagation method was 10 km based on idealized single plasma bubble setting. In this study, the space between each BP phase screen is 50 km. If shorten the distance between phase screens, would the estimation accuracy be enhanced? Also the authors mentioned that "in multiple bubble scenarios only the strongest disturbance would be resolved properly". How good would it be considered as "resolved properly"? If there're several local minima in the standard deviation curve, wouldn't different local minima correspond to locations of multiple irregularity regions?

Blekinge Institute of Technology
Department of Mathematics and Natural Sciences (TIMN)
Campus Gräsvik
371 79 Karlskrona

Telephone: +46 736 22 36 31
E-mail: vlb@bth.se
https://www.bth.se/timn

[Figure]

**Authors' reply:**
We agree that the BP method is not new, and we did not intend to present the method as a novelty. Despite the previous usage of the method to localize the irregularity regions, the accuracy of the estimate has not been thoroughly assessed, whichever were the limitations in previous studies. Since collocated cases are not broadly available, simulation cases can help investigate features observed in measurements. Ideally, the conclusions withdrawn from them still require validation by combined measurements.

On the accuracy stated in the manuscript, the BP amplitudes were calculated with a 10-km step. These amplitudes were plotted at a 50-km step (black vertical curves), so the figures were not crowded. Regardless, the numerical estimations reported in the manuscript are based on the 10-km resolution. We will make this information clearer in the document.

We performed an evaluation assuming shortened distances between planes in parallel to the influence of the SNR level on the estimation accuracy and precision. The evaluation considered the case reported in Carrano *et al.* (2011) and applied 20 different realizations of the plasma bubble – the same realizations reported in Fig. 8a.
Three additional distances between planes were considered: 1 km, 2.5 km and 5 km. For each one, two additional SNR scenarios were evaluated. Figure R1 depicts the SNR references assumed in the noise modelling.

[Figure]

Fig. R1: SNR references used during the accuracy and precision evaluation.

Blekinge Institute of Technology
Department of Mathematics and Natural Sciences (TIMN)
Campus Gräsvik
371 79 Karlskrona

Telephone: +46 736 22 36 31
E-mail: vlb@bth.se
https://www.bth.se/timn

[Figure]

Figure R2 shows the statistics for the location estimates obtained in the different scenarios.

[Figure]

Fig. R2: Estimated location for different PS resolutions and three SNR scenarios (i) original reference ($\sim 600\,\mathrm{V/V}$ at F-layer), constant at (ii) $700\,\mathrm{V/V}$ and (iii) $1000\,\mathrm{V/V}$. Each data point (asterisk) corresponds to the average BP estimation given the 20 realizations of the plasma bubble modelled in the forward propagation (see Figure 8a). The noise added to the signal was generated with the same seeds among the different SNR levels.

Blekinge Institute of Technology
Department of Mathematics and Natural Sciences (TIMN)
Campus Gräsvik
371 79 Karlskrona

Telephone: +46 736 22 36 31
E-mail: vlb@bth.se
https://www.bth.se/timn

It is important to mention that a 10-km phase screen (PS) resolution was used during the forward propagation. Despite the model setting the bubble at $Z_0 = -342.8\,\text{km}$, the nearest PS was placed at -346.7 km. Therefore, this position was considered the actual location during the analysis and is depicted as the red dashed line. This detail will be added to the manuscript.

Note that the data in the plots were discretized according to the PS resolutions. For example, the estimation obtained with 10-km resolution and original SNR (left panel). The estimated mean was $\bar{z} = -355\,\text{km}$ after 20 repetitions. Given the PS placement symmetric around the origin during the back propagation, the expected position was rounded to -360 km. Similarly, the computed confidence interval ($2\sigma$), depicted by the vertical bars, was $\pm 3.87\,\text{km}$. In this case, the confidence interval was not represented because the interval is smaller than half of the distance between PS.

In general, an improvement in accuracy is observed with higher resolutions. Such improvement is more noticeable when the original SNR reference is considered in the noise modelling. Further, the scenarios of higher SNR levels, without degradation in altitude, contributed to improving the precision of the estimations (shorter bars). These results and the discussion will be added to the revised manuscript.

Regarding the term "resolve", it may not be the most suitable one in this context since the BP method is used exclusively to estimate the horizontal distance at the current state. "Resolve" would be more appropriate if the shape of the plasma bubble was provided, as in tomography. The sentences containing the term "resolve" will be rephrased in the revised version of the manuscript. Nevertheless, the accuracy observed in scenarios presented in Fig. R2 is sufficient since it is smaller than the typical extension of irregularity patches, i.e., a few hundred kilometers.

In the context of multiple bubbles, we agree that the presence of multiple local minima likely corresponds to multiple irregularity regions – as modelled in simulations. However, this point has not been made clear in previous publications.

**Referee's comment**

The writing is poor, unclear and redundant. For instance,

L47-48: "the location of irregularities patches is not self-reliant": what does "self-reliant" mean herein?

L111-113: "the excess path due to ionospheric ... which ... which ... due to slightly different propagation paths": please rephrase this sentence.

L118-120: "Such regions ..., which specifically corresponds to sizes up to the Fresnel scale": I can't tell what you are trying to express here, especially the clause.

L131-132: "The high-order bias ... by Kappa or Bi-local correction": This sentence is not needed.

I would stop addressing the remaining ones here, and suggest the authors read the whole manuscript carefully and try to make each statement clearer and more concise.

Blekinge Institute of Technology      Telephone: +46 736 22 36 31
Department of Mathematics and Natural Sciences (TIMN)      E-mail: vlb@bth.se
Campus Gräsvik      https://www.bth.se/timn
371 79 Karlskrona

[Figure]

**Authors' reply:**
These comments/suggestions are going to be covered in the revised manuscript.

**Referee's comment**
L149: "outer scale": what does this "outer scale" indicate?

**Authors' reply:**
The outer scale is a parameter of the power law function modelling the plasma bubble during forward propagation. It adds a cutoff point at the low-end region of the power spectral density and relates to the largest irregularity scale size that creates turbulence in the signal. The Fresnel filtering generally suppresses this cutoff in the intensity spectrum. The term "outer scale" was introduced in the context of weak scattering in the 70s:

- Dyson, P. L., McClure, J. P., Hanson, W. B., "In situ measurements of the spectral characteristics of F region ionospheric irregularities," *Journal of Geophysical Research*, 79(10), 1497–1502, Apr. 1974. doi: 10.1029/ja079i010p01497

- Yeh, K. C., Liu, C. H., Youakim, M. Y., "A theoretical study of the ionospheric scintillation behavior caused by multiple scattering," *Radio Science*, 10(1), 97–106, Jan. 1975. doi:10.1029/RS010i001p00097

- Rino, C. L., Livingston, R. C., Whitney, H. E., "Some new results on the statistics of radio wave scintillation, 1. empirical evidence for gaussian statistics," *Journal of Geophysical Research*, 81(13), 2051–2057, May 1976. doi:10.1029/ja081i013p02051

**Referee's comment**
L166-167: Could the authors please give a simple explanation of your WOP or tell me where in the text you introduced your "WOP"?

Blekinge Institute of Technology
Department of Mathematics and Natural Sciences (TIMN)
Campus Gräsvik
371 79 Karlskrona

Telephone: +46 736 22 36 31
E-mail: vlb@bth.se
https://www.bth.se/timn

**Authors' reply:**
In the submitted manuscript, we have only cited Knepp (1983) as a reference to the wave optics propagator (WOP), specifically the multiple phase screen (MPS). The focus of the manuscript is not the WOP implementation. Since Knepp's reference is rather dated, we will add references to our previous publication, where the implementation is described in more detail.

- V. Ludwig-Barbosa, J. Rasch, A. Carlström, M. I. Pettersson, and V. T. Vu, "GNSS Radio Occultation Simulation Using Multiple Phase Screen Orbit Sampling," *IEEE Geosci. Remote Sens. Lett.*, vol. 17, no. 8, pp. 1323–1327, Aug. 2020. doi:10.1109/LGRS.2019.2944537.

- V. Ludwig-Barbosa, T. Sievert, J. Rasch, A. Carlström, M. I. Pettersson, and V. T. Vu, "Evaluation of Ionospheric Scintillation in GNSS Radio Occultation Measurements and Simulations," *Radio Sci.*, vol. 55, no. 8, Aug. 2020. doi:10.1029/2019RS006996.

**Referee's comment**
`L219-220`: Please explain where the threshold value comes from. There's no Figure A.

**Authors' reply:**
The threshold is given after the amplitude of the estimated noise level around the F-layer density peak ($hmF2$). The calculation of such value is described in Appendix A. The noise amplitude is depicted in Fig. A1(b) as a function of SLTA. The reference for the figure was previously incomplete. It has been corrected.

**General comment:**
"Specific comments" not listed in this document will be addressed and modified in the updated manuscript.

Blekinge Institute of Technology
Department of Mathematics and Natural Sciences (TIMN)
Campus Gräsvik
371 79 Karlskrona

Telephone: +46 736 22 36 31
E-mail: vlb@bth.se
https://www.bth.se/timn

---

## Author Comment (AC2)

**Editorial Board and Referee #2**
**Atmospheric Measurement Techniques, EGU**

**Manuscript ID:**
AMT-2022-57

**Correspondent:**
Vinícius Ludwig Barbosa
Department of Mathematics and Natural Sciences
Blekinge Institute of Technology
Campus Gräsvik, 371 79 Karlskrona

**Dear Referee #2,**

Thanks for your comments on AMT-2022-57 "Detection and Localization of F-layer Ionospheric Irregularities with Back Propagation Method Along Radio Occultation Ray Path". The authors have considered them, and this document contains our answers.

An updated manuscript with modifications based on the comments of both reviewers will be made available soon.

Best regards,
Authors

Blekinge Institute of Technology
Department of Mathematics and Natural Sciences (TIMN)
Campus Gräsvik
371 79 Karlskrona

Telephone: +46 736 22 36 31
e-mail: vlb@bth.se
https://www.bth.se/timn

[Figure]

**Referee's comment**
This paper investigates the detection and localization of F-layer ionospheric irregularities with the back propagation (BP) method.

The most confusing me is that the authors listed the observations of the sporadic E layers from two COSMIC RO measurements in the 4.3 Analysis COSMIC occultations results, while Section 3 is the simulation of plasma bubbles in the F-region with the BP method. They even did not analyze the sensitivity level and estimation accuracy of sporadic E-layers since it is beyond the scope of this study (line 238). What is section 4.3 meant to explain? This may indicate that two or more irregularity regions occurred in the E region, but not the F-layer ionospheric irregularities.

I suggest the authors analyze the F-layer ionospheric irregularities from the COSMIC RO measurements and compare the simulations and observations to validate the detection and localization estimate with BP method along radio occultation ray path. Otherwise, conclusions made here are not very convincing through only simulations. That is my main comment.

> **Authors' reply:**
> Fig. R1 shows the SNR for the two COSMIC occultation events discussed in Section 4.3.

[Figure]

(a)             (b)

Fig. R1: COSMIC occultation events in DOY 174, 2015 presented in Section 4.3: SNR (black) and $S_4$ index (red) in different tangent heights.

Blekinge Institute of Technology
Department of Mathematics and Natural Sciences (TIMN)
Campus Gräsvik
371 79 Karlskrona

Telephone: +46 736 22 36 31
e-mail: vlb@bth.se
https://www.bth.se/timn

[Figure]

The SNR in Fig. R1(a) does not give a clear indication of scintillation due to sporadic-Es, either by containing one or multiple u-shape fades as presented in the following reference:

- Yue, X., Schreiner, W. S., Zeng, Z., Kuo, Y.-H., Xue, X., "Case study on complex sporadic E layers observed by GPS radio occultations," *Atmospheric Measurement Techniques*, 8(1), 225–236, 2015. https://doi.org/10.5194/amt-8-225-2015.

The disturbance seems to have a stable variance and its overall pattern is similar to the cases reported in:

- Wickert, J., Pavelyev, A. G., Liou, Y. A., Schmidt, T., Reigber, C., Igarashi, K., Pavelyev, A. A., Matyugov, S. S., "Amplitude variations in GPS signals as a possible indicator of ionospheric structures," *Geophysical Research Letters*, 31(24), L24801, 2004. https://doi.org/10.1029/2004GL020607¡/div¿

However, the characteristics of a sporadic-E scintillation case are observed in Fig. R1(b). An u-shape fade is observed around 100 km SLTA. This altitude is within the common range for sporadic-E occurrences, increasing the likelihood that the disturbance layer is aligned with the propagation direction (Zeng et al., 2010). The $S_4$ index value is considerably stable outside the SLTA range affected by the u-shaped fade and, therefore, presumably unrelated to the sporadic-Es.

In Fig. 15, the detrended BP amplitude standard deviation (red curve) was computed outside the altitude range for sporadic Es. Therefore, we reduced the chances of having the estimation contaminated by sporadic-E irregularities. The same methodology was described by Cherniak et al. (2019). We will add this discussion to the revised manuscript.

The results of the BP method in simulations assuming two irregularity regions in the F-region (Fig. 11, 12, 14) show $\sigma_u$ curves resembling the pattern observed in BP results of the two COSMIC occultation events. Thus, the hypothesis of disturbances observed outside the sporadic-E regions being caused by F-region irregularities is plausible in these measurements.

Nevertheless, we would like to stress the importance of collocating occultation events to other techniques in future studies. In this manuscript, we have used occultation events previously investigated in publications applying the BP method.

Blekinge Institute of Technology
Department of Mathematics and Natural Sciences (TIMN)
Campus Gräsvik
371 79 Karlskrona

Telephone: +46 736 22 36 31
e-mail: vlb@bth.se
https://www.bth.se/timn

[Figure]

**Referee's comment**

The descriptions about back propagation are too brief and incomplete in the text. Some recent similar work should also be referenced and mentioned. Figures 1 and 2 (but not limited to) are difficult for the reader to understand without detailed captions. Please add appropriate content and improve the diagram to make it easier for readers to read.

> **Authors' reply:**
>
> Following your recommendation, the section describing the BP method has been modified. We hope the changes are satisfactory and have made the text more comprehensive.
>
> Next, a search was made to collect related publications in the platform Scopus using the following search string:
>
> ```
> ALL(("backpropagation" OR "back-propagation" OR "back
> propagation") AND ("radio occultation") AND (ionospher*))
> ```
>
> All references containing the theoretical definitions for the method are already cited in the manuscript. Newer publications focus on the method's application and not on the theoretical background. Among these, the following references will be added in the upcoming manuscript version:
>
> - Carrano, C. S., Groves, K. M., Mcneil, W. J., Doherty, P. H. (2012). "Scintillation Characteristics across the GPS Frequency Band," *25th International Technical Meeting of the Satellite Division of The Institute of Navigation (ION GNSS)*, 1972–1989, Nashville, TN, September 2012.
>
> - Carrano, C. S., Groves, K. M., Delay, S. H., Doherty, P. H., "An inverse diffraction technique for scaling measurements of ionospheric scintillations on the GPS L1, L2, and L5 carriers to other frequencies," *Institute of Navigation International Technical Meeting (ITM)*, 709–719, San Diego, CA, January 2014.
>
> - Sokolovskiy, S., Schreiner, W. S., Zeng, Z., Hunt, D. C., Kuo, Y.-H., Meehan, T. K., Stecheson, T. W., Mannucci, A. J., & Ao, C. O., "Use of the L2C signal for inversions of GPS radio occultation data in the neutral atmosphere," *GPS Solutions*, 18(3), 405–416, 2014. https://doi.org/10.1007/s10291-013-0340-x
>
> Finally, we have made minor modifications to the diagrams shown in Fig. 1 and 2, and rewrote their captions.

**Referee's comment**

Please add appropriate text captions in Figures 6 and 7.
* * *
Blekinge Institute of Technology
Department of Mathematics and Natural Sciences (TIMN)
Campus Gräsvik
371 79 Karlskrona

Telephone: +46 736 22 36 31
e-mail: vlb@bth.se
https://www.bth.se/timn

[Figure]

**Authors' reply:**
The text captions have been rephrased to describe better these figures and others introduced at a later point in the document.

Blekinge Institute of Technology
Department of Mathematics and Natural Sciences (TIMN)
Campus Gräsvik
371 79 Karlskrona

Telephone: +46 736 22 36 31
e-mail: vlb@bth.se
https://www.bth.se/timn

[Figure]

**Referee's minor comments**

`Lines 41-42`: Please reexamine carefully, it seems difficult to derive this description from the literature you cited.

We have added the following reference to complement the initial citation:

- Cherniak, I. and Zakharenkova, I., "High-latitude ionospheric irregularities: differences between ground- and space-based GPS measurements during the 2015 St. Patrick's Day storm," *Earth, Planets and Space*, 68(136), 2016. https://doi.org/10.1186/s40623-016-0506-1

`Figure 6`: The last number on the Y-axis, 0, is half missing. Other pictures have similar problems. Please check it.

We have updated all figures showing simulation results. The overall quality of the figures has been improved.

`Lines 222-223`: I don't know how did you arrive at this "$\sigma_{\Delta\rho/\rho} = 3.0\%$ represents $S_4 < 0.1$". In addition, please show the reference or basis for the low scintillation threshold ($S_4 = 0.2$).

The $S_4$ index value was calculated at the LEO orbit plane after the forward propagation assuming $\sigma_{\Delta\rho/\rho} = 3.0\%$. We will add the following references to the manuscript to motivate the threshold $S_4 \leq 0.2$:

- Béniguel, Y., Romano, V., Alfonsi, L., Aquino, M., Bourdillon, A., Cannon, P., Franceschi, G. de, Dubey, S., Forte, B., Gherm, V., Jakowski, N., Materassi, M., Noack, T., Pozoga, M., Rogers, N., Spalla, P., Strangeways, H. J., Warrington, E. M., Wernik, A. W., Zernov, N. (2009). Ionospheric scintillation monitoring and modelling. *Annals of Geophysics*, 391–416. https://doi.org/10.4401/ag-4595;

- Ma, G., Hocke, K., Li, J., Wan, Q., Lu, W., Fu, W. (2019). GNSS Ionosphere Sounding of Equatorial Plasma Bubbles. *Atmosphere*, 10(676), 1–11. https://doi.org/10.3390/atmos10110676.

Below such a threshold, disturbances produced by other sources can be similar to weak ionospheric scintillation.
* * *
Blekinge Institute of Technology    Telephone: +46 736 22 36 31
Department of Mathematics and Natural Sciences (TIMN)    e-mail: vlb@bth.se
Campus Gräsvik    https://www.bth.se/timn
371 79 Karlskrona

[Figure]

`Line 242`: by what threshold values are the irregularities still detectable?

We establish that irregularities are detectable as long as $\sigma_0 = 0.0456$, defined after the receiver noise level considered in our study, i.e., $\sigma_u > \sigma_0$. Nevertheless, the receiver noise of each system bounds this threshold. This statement will be made more explicit in the updated manuscript.

**General comment:**
Comments not listed in the document will be addressed in the revised manuscript.

Blekinge Institute of Technology

Department of Mathematics and Natural Sciences (TIMN)

Campus Gräsvik

371 79 Karlskrona

Telephone: +46 736 22 36 31

e-mail: vlb@bth.se

https://www.bth.se/timn

---

## Referee Report (RR1)

Referee report on "Detection and Localization of F-layer Ionospheric Irregularities with Back Propagation Method Along Radio Occultation Ray Path", by Ludwig-Barbosa et al.

The authors have addressed my comments. They show that the S4 index value is considerably stable outside the SLTA range affected by the soiraduc E sporadic-E scintillation (Zeng and Sokolovskiy, 2010; Wickert et al., 2004; Arras and Wickert, 2018), and therefore they reduced the chances of having the estimation contaminated by sporadic-E irregularities. Besides, some recent references should also be referenced:

Resende, L. C. A., Arras, C., Batista, I. S., Denardini, C. M., Bertollotto, T. O., & Moro, J. (2018). Study of sporadic E layers based on GPS radio occultation measurements and Digisonde data over the Brazilian region. Annales Geophysicae, 36, 587–593.

Yu, B., Xue, X., Yue, X., Yang, C., Yu, C., Dou, X., et al. (2019). The global climatology of the intensity of the ionospheric sporadic E layer. Atmospheric Chemistry and Physics, 19(6), 4139–4151.

Yu, B.; Scott, C.J.; Xue, X.; Yue, X.; Dou, X. Derivation of global ionospheric Sporadic E critical frequency (fo Es) data from the amplitude variations in GPS/GNSS radio occultations. R. Soc. Open Sci. 2020, 7, 200320.

Carmona, R. A., Nava, O. A., Dao, E. V., & Emmons, D. J. (2022). A comparison of sporadic-E occurrence rates using GPS radio occultation and ionosonde measurements. Remote Sensing, 14(3), 581.

**Minor comments**

1. Abstract: Line 15 provide insight into…

2. Line 238 Different from…

3. Conclusions: Line 395: . Approaches to estimate such features…

---

## Author Response (AR2)

**Editorial Board and Referee**
**Atmospheric Measurement Techniques, EGU**

**Manuscript ID:**
AMT-2022-57

**Correspondent:**
Vinícius Ludwig Barbosa
Department of Mathematics and Natural Sciences
Blekinge Institute of Technology
Campus Gräsvik, 371 79 Karlskrona

**Dear Referee,**

Thanks for your additional comments on AMT-2022-57 "Detection and Localization of F-layer Ionospheric Irregularities with Back Propagation Method Along Radio Occultation Ray Path".

The authors have added the suggested references (4) and completed the three minor changes in the text.

Finally, a couple of abbreviations have been added in the abstract and in the rest of the document – following the recommendation from the Associate Editor.

Best regards,
Authors